# Pangenome analysis of transposable element insertion polymorphisms reveals features underlying cold tolerance in rice

Yongqing Qian[1,5], Zuwen Zhou [2,5], Tianmin Ouyang[2,5], Dongao Li[2], Ru Li[2], Ping Gan[2], Renfei Qiao[2], Yingying Tan[1], Mingchao Qian[1], Liezhao Liu [1], Jiana Li [1], Kun Lu [1,3] ✉, Jijing Luo [2] ✉, Ling-Ling Chen [2,4] ✉ & Jia-Ming Song [1,3] ✉

Transposable elements (TEs) introduce genetic and epigenetic variability, contributing to gene expression patterns that drive adaptive evolution in plants. Here, we investigate TE architecture and its effect on cold tolerance in rice. By analyzing a pangenome graph and the resequencing data of 165 rice accessions, we identify 30,316 transposable element insertion polymorphism (TIP) sites, highlighting significant diversity among polymorphic TEs (pTEs). We observe that pTEs exhibit increased H3K27me3 enrichment, suggesting a potential role in epigenetic differentiation under cold stress and in the transcriptional regulation of the cold response. We identify 26,914 TEs responsive to cold stress from transcriptome data, indicating their potential significance in regulatory networks for this response. Our TIP-GWAS analysis reveal two cold tolerance genes *OsCACT* and *OsPTR*. The biological functions of these genes are confirmed using knockout and overexpression lines. Our web tool (https://cbi.gxu.edu.cn/RICEPTEDB/) makes all pTEs available to researchers for further analysis. These findings provide valuable targets for breeding cold-tolerant rice varieties, indicating the potential importance of pTEs in crop enhancement.

Rice (*Oryza sativa* L.) comprises two major subspecies, *indica* and *japonica*[1]. Rice is one of the most important staple crops worldwide since it sustains nearly half the global population[1–3]. The critical role of rice in global food security is unquestionable; however, rice yields are significantly affected by various environmental stresses, including natural disasters[4,5] and cold damage[6]. To address this challenge, molecular genetic tools have been utilized to increase the cold tolerance of rice[7]. Advances in understanding the mechanisms underlying cold resistance and transcriptional regulation have identified key players, such as *COLD1* and *OsCNGC9*, which activate calcium signaling pathways in response to cold stress[7,8]. Additionally, *COLD11* has been shown to repair DNA damage induced by cold stress[9], whereas *OsRS2Z38* modulates cold tolerance through alternative splicing[10]. The discovery of these cold tolerance genes and their mechanisms opens new avenues for modern molecular breeding, facilitating the development of rice varieties with improved resistance to cold.

Importantly, genomic research is shifting away from reliance on a single reference genome to a new era of graph-based pangenomes,

[1]Integrative Science Center of Germplasm Creation in Western China (CHONGQING) Science City and Southwest University, College of Agronomy and Biotechnology, Southwest University, Chongqing, China. [2]State Key Laboratory for Conservation and Utilization of Subtropical Agro-bioresources, College of Life Science and Technology, Guangxi University, Nanning, China. [3]Engineering Research Center of South Upland Agriculture, Ministry of Education, Chongqing, China. [4]Yazhouwan National Laboratory, Sanya, China. [5]These authors contributed equally: Yongqing Qian, Zuwen Zhou, Tianmin Ouyang. ✉e-mail: drlukun@swu.edu.cn; jjluo@gxu.edu.cn; llchen@gxu.edu.cn; jmsong@swu.edu.cn

which store and display sequence variation in species[11,12]. By leveraging the ability of pangenomes to reveal structural variations (SVs), researchers have identified novel functional genes[13–16]. These SVs play crucial roles in plant genomes, influencing gene structure, histone modifications, and the expression of long noncoding RNAs (lncRNAs) or proteins that regulate gene activity[17–19]. Many of these SVs are driven by transposable elements (TEs)[20–24].

Recent evidence suggests that rice TEs may provide further insights into the regulation of cold tolerance, allowing the identification of additional cold-tolerance genes and broadening the pool of breeding targets[7]. Accumulating evidence has associated TEs with plant phenotypic variation and evolution since their discovery by Barbara McClintock[25]. A well-known example includes long terminal repeat retrotransposons (LTR-RTs), such as *Hopscotch* in maize, which functions as an enhancer of the *tb1* gene, contributing to apical dominance[26]. TEs are grouped into two classes on the basis of their transposition mechanisms: retrotransposons and DNA transposons. Retrotransposons are further divided into those with long terminal repeats (LTRs), such as the *Gypsy* and *Copia* families, and non-LTR elements, including long interspersed nuclear elements (LINEs) and short interspersed nuclear elements (SINEs). DNA transposons are classified into various families, including miniature inverted repeat transposable elements (MITEs), *CACTA*, *hAT*, *Tc1-Mariner*, *Mutator*, *PIF-Harbinger*, and *Helitrons*[27]. In rice, the DNA transposon *mPing* regulates gene networks involved in stress responses[28–31]. Furthermore, research indicates that transposable element insertion polymorphisms (TIPs) exert a greater influence on phenotypic traits than do single-nucleotide polymorphisms (SNPs)[1,32–35]. Recent studies focused on identifying TEs in rice populations and constructing pan-TE maps have revealed numerous TE characteristics associated with rice domestication and agronomic traits[36,37]. However, the influence of TEs on cold tolerance in rice remains largely unexplored.

In this study, we conduct de novo genome assembly of 10 rice lines utilizing Oxford Nanopore Technologies (ONT) and Illumina sequencing technologies. Combining with the previously published genome of *indica* rice MH63[38], we construct a pangenome graph and identify 30,316 TIP sites, highlighting the diversity of polymorphic TEs (pTEs) in the rice genome. Using transcriptome data, we analyse how pTEs contribute to differential gene expression prior to and following cold stress. Furthermore, we integrate epigenetic modification data to elucidate the mechanisms through which pTEs influence gene regulation. In addition, we identify numerous cold-responsive genes and pTEs, and conduct co-expression analysis with known cold tolerance genes and pTEs, identifying pTEs that potentially encode lncRNAs. Our TIP-GWAS analysis of cold tolerance phenotypes in 165 rice accessions leads to the discovery of cold tolerance genes, such as *OsCACT*. Furthermore, overexpression experiments and metabolomic data reveal that *OsCACT* enhances cold tolerance by regulating fatty acid metabolism and antioxidant activity during cold stress. These findings provide potential target genes for breeding cold-tolerant rice varieties.

## Results

### De novo assembly and annotation of 10 rice accessions

We selected 10 rice accessions, including 6 *indica* varieties and 4 *japonica* varieties, that are geographically diverse and represent a broad spectrum of temperature adaptability in rice (Supplementary Fig. 1 and Supplementary Table 1). Using 242.0 Gb (~62x) from ONT long read sequencing and 213.7 Gb (~55x) from Illumina short read sequencing (Supplementary Table 2), we assembled 10 high-quality rice genomes (see Methods). The assembled genome sizes ranged from 373 to 394 Mb (Fig. 1a), closely matching the telomere-to-telomere (T2T) rice reference genome[38,39]. We evaluated the assembly quality using multiple metrics (Table 1 and Supplementary Table 3). The average quality value (QV) was approximately 35, indicating high assembly accuracy[40]. The LTR assembly index (LAI) values, which are

used to evaluate the integrity of intact LTRs in these assemblies, were all above 20, achieving 'gold-standard' quality[41]. The assembly quality indicator (AQI) score ranged from 97.90 to 100, reflecting reference-level quality[42], while the average mapping rate of Illumina reads was 99.53%. We also utilized benchmarking universal single-copy orthologues (BUSCO)[43] to evaluate the completeness of each assembly, and found an average BUSCO completeness of 98.7%. Furthermore, we predicted nearly complete telomeric and centromeric regions (Fig. 1a) for each assembly (Table 1 and Supplementary Fig. 2), with successful assembly of several gap-free chromosomes (Supplementary Table 3).

We performed a synteny analysis between our 10 assemblies and the gap-free rice genome MH63[38] (Supplementary Fig. 3). All 6 *indica* genomes exhibited strong synteny with MH63, whereas *japonica* genomes displayed notable *indica*–*japonica* differentiation, particularly on chromosome 6. To validate the authenticity of the inversion observed in this region, we examined read coverage across the inversion breakpoint (Supplementary Fig. 4) and found no evidence of breaks. Additionally, the genome synteny between MH63 (*indica*)[38] and Nipponbare (*japonica*)[39] was highly similar (Supplementary Fig. 3), further supporting the authenticity of the inversion. To characterize the genomic differences, we utilized MH63 as a reference to identify chromosome rearrangements among the genomes (Supplementary Table 4). Excluding variations caused by gaps, we observed high concordance in SVs between our assembled Nipponbare genome and the T2T Nipponbare ref. [39].

We annotated TEs across 10 genomes using Extensive de novo TE Annotator (EDTA)[44], identifying an average of 204.6 Mb of TEs per assembly, accounting for 51.91% to 54.05% of the total assembly length (Fig. 1b and Supplementary Table 5), which was similar to previously reported rice genomes[45,46]. These TEs were classified into two primary classes: retrotransposons, which constituted 22.24% to 25.72% of the genome, and DNA transposons, comprising 27.60% to 29.10%. *Gypsy* elements, the most abundant TE type, accounted for 16.29% to 20.27% of the genome and were the primary contributors to overall TE variation among the genomes[47]. On average, we identified 21,768 full-length LTR-RTs, with a combined length of approximately 56.5 Mb (Supplementary Table 5). Among the two most abundant superfamilies *Gypsy* and *Helitron*, we found that *Gypsy* elements were enriched near centromeres, suggesting potential involvement in centromeric functions, whereas *Helitron*s were distributed almost uniformly across the genome (Fig. 1a).

We conducted homology-based annotation of protein-coding genes across the 10 genomes and predicted between 28,878 and 33,690 protein-coding genes (Table 1). Among these annotated genes, between 25,779 and 30,132 contained TE insertions within their 2 kb upstream regions, whereas between 24,156 and 28,364 contained insertions in the 2 kb downstream regions. Additionally, between 12,120 and 14,472 TEs were inserted into introns, whereas between 9698 and 12,535 were inserted into coding regions. Overall, an average of 32.69% of the TEs were located within 2 kb of genes, suggesting that these TEs, which are expanded and inserted near rice genes, may influence gene expression regulation. On the basis of these protein-coding genes, we identified 30,327 rice pangene families, including 18,979 core gene families, accounting for approximately 62.6% of the total pangene families (Fig. 1c), which is consistent with a previous study[48]. In summary, the 10 genomes exhibited high accuracy, continuity, and genomic integrity, providing valuable genomic resources for subsequent pangenome analysis.

### Construction and analysis of a TIP map based on the pangenome graph

We constructed a rice pangenome graph by integrating the MH63 genome with our 10 assemblies using minigraph[49]. The total size of the pangenome graph was 581.7 Mb and contained 50,875 SVs, which included 14,989 insertions, 10,834 deletions, and 25,052 allelic

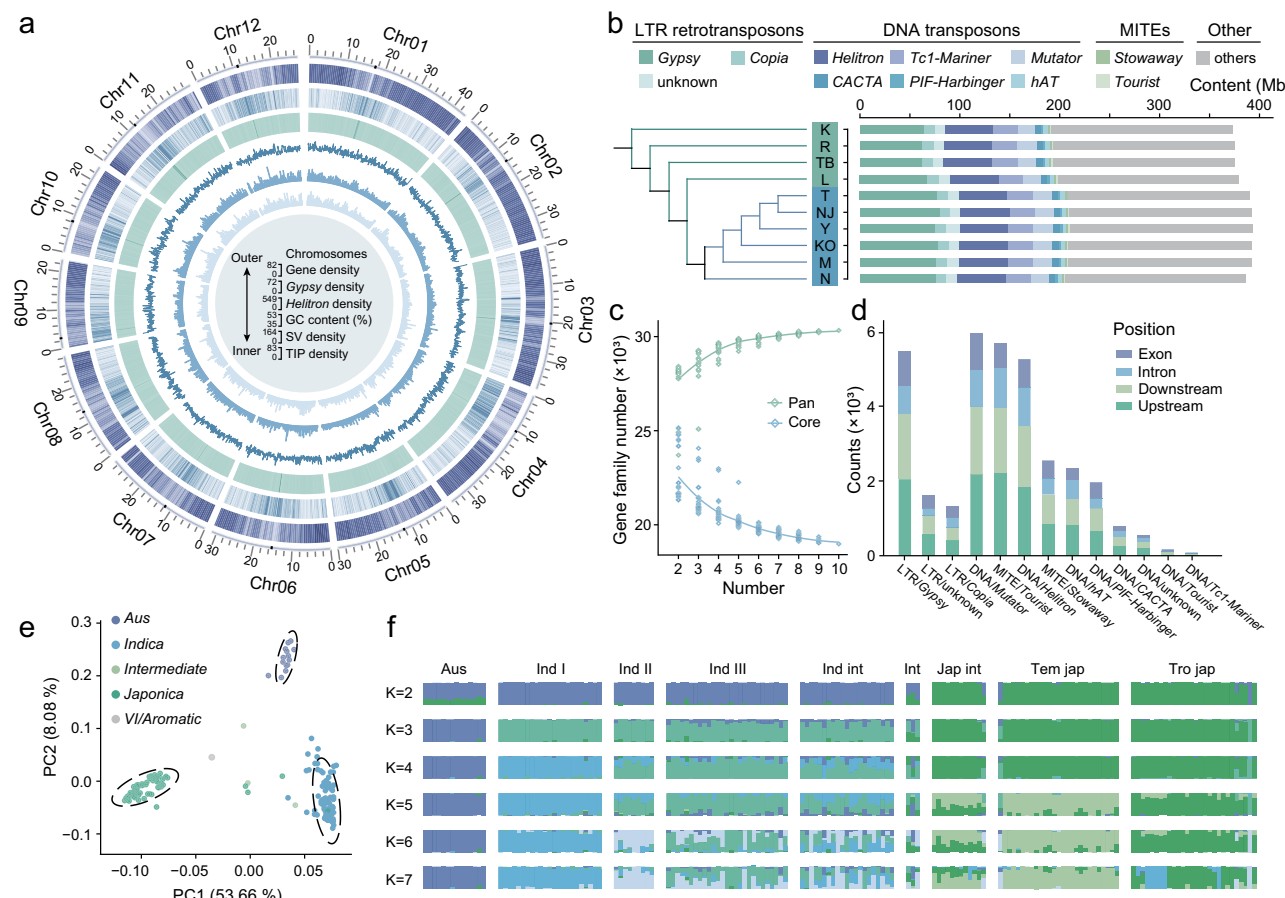

**Fig. 1 | Genomic features and population structure of 165 rice accessions.** **a** Circos plot illustrating the genomic characteristics of the TB accession. The outer to inner circles represent: chromosomes (with centromeres highlighted in dark blue), gene density, *Gypsy* density, *Helitron* density, GC content, structural variation (SV) density, and TIP density. The shades transitioning from light to dark indicate increasing numerical values. The numbers labeling the outer circle indicate the length (Mb) of each chromosome. **b** TE content across the 10 rice assemblies. The phylogenetic tree was constructed based on single-copy genes from the assemblies. **c** Variation in gene families within the pan-genome and core genome as the number of rice genomes increases. **d** The number of different TE insertions in genic regions, which include sequences 2 kb upstream and downstream of each gene body. **e** Principal component analysis (PCA) plot of 165 rice accessions based on TIPs. **f** Population structure analysis conducted for 165 accessions with varying ancestry kinship (K = 2–7) based on TIPs. Each vertical bar represents an accession, and the x-axis displays the different subpopulations[145]: Aus (*aus*), Ind I (*indica* I), Ind II (*indica* II), Ind III (*indica* III), Ind int (*indica* intermediate), Int (intermediate), Jap int (*japonica* intermediate), Tem jap (temperate *japonica*) and Tro jap (tropical *japonica*). The y-axis quantifies ancestry membership. Source data are provided as a Source Data file.

**Table 1 | Summary statistics of 10 rice genomes**

| Accession | Genome size (bp) | Contig N50 (Mb) | Telomere number | Centromere number | Completeness (BUSCO) | LAI | QV | Gene number |
|---|---|---|---|---|---|---|---|---|
| KOGONI | 391,943,762 | 11 | 12 | 12 | 98.70% | 24.27 | 35.4 | 33553 |
| Koshihikari | 373,101,744 | 9 | 13 | 12 | 98.60% | 21.41 | 34.8 | 28878 |
| Lemont | 379,430,320 | 25 | 12 | 12 | 98.60% | 22.32 | 35.3 | 29369 |
| MADINIKA | 392,020,413 | 21 | 13 | 12 | 98.80% | 24.22 | 35.7 | 32761 |
| Nanjing11 | 392,195,977 | 16 | 9 | 12 | 98.60% | 25.14 | 35.3 | 33024 |
| NONA_BOKRA | 386,246,295 | 15 | 14 | 11 | 98.50% | 23.58 | 35.6 | 32382 |
| Nipponbare | 374,980,003 | 15 | 9 | 12 | 98.70% | 21.43 | 35 | 28936 |
| Fujisaka5 | 375,578,160 | 21 | 17 | 12 | 98.80% | 21.67 | 35.6 | 29124 |
| TEQING | 390,364,816 | 14 | 13 | 12 | 98.70% | 23.96 | 34.9 | 33196 |
| 9311 | 393,694,398 | 16 | 14 | 12 | 98.60% | 24.02 | 34.8 | 33690 |

variations, with allele counts ranging from 2 to 7 (Supplementary Fig. 5). Among these allelic variations, 45,066 (88.6%) were biallelic variants, whereas 9,194 were conserved between *indica* and *japonica*, indicating a substantial number of variety-specific SVs beyond those that are conserved. We then annotated and merged the TE sequences from these 10 assemblies to construct a rice pan-TE set[50]. The pan-TEs

totaled 267.7 Mb and comprised 142,909 sequences. By annotating SVs with pan-TE sequences and applying the 80–80 rule[27], we identified 31,673 SVs containing TEs. We further annotated these SVs using EDTA[44]. The total length of these sequences was 118.5 Mb, with 95.1 Mb (80.29%) annotated as TEs, most of which were *Gypsy* elements, accounting for 49.17% of the total length. In total, 30,316 SVs were

identified as TIP sites. We randomly selected 8 TIPs and validated their breakpoints through long-read mapping, which provided independent support for the accuracy of the identified TIPs (Supplementary Fig. 6). These findings further emphasize the role of TEs as major drivers of SVs.

Overall, the pattern of pTE insertions in these assemblies (Supplementary Fig. 7) was similar to the trends observed in tomato[34], suggesting that most pTE insertions occurred after the divergence of subspecies. However, pTE insertion rates were higher in the 4 assemblies (Supplementary Fig. 7), with the peak contributing primarily to TEs shared among the 4 *japonica* assemblies. These results indicate that many pTEs were fixed after *indica–japonica* divergence and played a key role in maintaining their divergence. Additionally, we found that the distributions of gene-flanking pTEs, genome-wide pTEs, and genome-wide SVs were nearly identical (Supplementary Fig. 7), suggesting that both pTEs and SVs were subjected to similar selective pressures during rice domestication.

To explore the relationship between pTEs and gene locations, we defined TIP genes, i.e., gene bodies that were within 2 kb upstream or downstream of a pTE. In the rice pangenome graph, we identified 15,440 TIP genes, indicating that approximately 39% of the genes may be influenced by pTEs. Given the potential for multiple pTEs near a single gene or a single pTE near multiple genes, we categorized each occurrence of a TE within 2 kb of a gene as an 'event'. In total, we identified 24,376 such events in the rice pangenome graph. Among these events, 7559 occurred within gene bodies, with 3201 located in exons and 4358 in introns. Additionally, 9131 events were identified within 2 kb upstream (promoter regions) of genes, and 7686 events were identified within 2 kb downstream of genes. In general, the pTE insertion densities indicated a preference for insertion in the upstream 2 kb region (Supplementary Fig. 8), potentially influencing gene expression through the regulation of promoter activity. With respect to TE types, approximately 25% of the pTEs were retrotransposons, whereas 75% were DNA transposons, with considerable variation in both their numbers and insertion locations (Fig. 1d and Supplementary Fig. 8). MITE/*Tourist*-type pTEs exhibited a strong preference for insertion in the 2 kb regions flanking genes rather than within gene bodies (Supplementary Fig. 8). Previous studies have also reported that MITEs tend to persist in upstream gene regions[28,51], suggesting that MITE elements are more likely to function as regulatory elements in noncoding regions[52,53].

To explore the potential value of pTEs in rice population genetics, we selected 165 rice accessions from the 3k Rice Genome Project[1] (Supplementary Data 1) and conducted population analyses using both TIPs and SNPs. Principal component analysis (PCA) revealed that these rice varieties clearly clustered into three distinct subpopulations, representing *japonica*, *indica*, and *Aus* (Fig. 1e and Supplementary Fig. 9a). Population structure analysis revealed that *japonica*, *indica*, and *Aus* were the first three groups to separate, with further differentiation observed as the value of K increased (Fig. 1f and Supplementary Fig. 9b). At K = 5, we observed further subdivisions within *indica* (*Indica* I) and *japonica* (temperate and tropical *japonica*) varieties, reflecting more detailed population stratification. These findings indicate that pTEs can accurately delineate rice population structure, which is consistent with the SNP results. Moreover, the differences in pTE insertion sites and the conserved patterns between subspecies suggest that pTEs are closely associated with gene expression and species differentiation.

## Transcriptional regulation and epigenetic profiles of pTEs

We constructed a rice TIP map following the pipeline outlined in Fig. 2a and released a web tool (https://cbi.gxu.edu.cn/RICEPTEDB/) that makes 30,316 pTEs available to researchers. To broaden the scope and applicability of our study, we collected 131 published high-quality rice genomes[38,39,45,48,54,55] and combined them with our 10 assembled genomes to construct an expanded pan-TE dataset, which is also available through our web tool (https://cbi.gxu.edu.cn/RICEPTEDB/).

To further explore the potential regulatory functions of pTEs, we conducted RNA-seq analysis on the 10 rice accessions under three conditions: control (0 h), 24-hour (24 h) cold treatment, and 72-hour (72 h) cold treatment (Supplementary Data 2), to assess the impact of pTEs on gene expression. Overall, we observed significant differences across all conditions in the expression levels of TIP genes when comparing accessions with pTE insertions to those without (Supplementary Fig. 10). Furthermore, various types of pTEs exhibited distinct effects on gene expression (Supplementary Fig. 10), with DNA/*Mutator*, MITE/*Tourist*, DNA/*Helitron*, and DNA/*PIF-Harbinger* demonstrating particularly significant impacts following cold treatment. These findings suggest that these TE types are likely to play a role in regulating cold tolerance mechanisms in rice. A well-known example is the MITE/*Tourist* element *mPing*, which confers stress-inducible expression on nearby genes[29,30]. Moreover, the location of pTE insertions influences gene expression. To isolate the effect of pTE insertion sites as a single variable, we analysed genes with unique pTE insertions in the region 2 kb upstream, the region 2 kb downstream, exons, and introns of genes, resulting in 4573, 3827, 1584, and 1385 genes, respectively (Fig. 2b). We found that pTE insertions within the upstream 2 kb and intronic regions were significantly associated with lower gene expression across all time points, whereas insertions in exons and downstream 2 kb regions were not significantly associated with expression changes (Fig. 2c and Supplementary Fig. 11). In conclusion, pTEs are broadly associated with gene expression in a category- and site-specific manner, with certain TE types and locations exerting a more profound impact, especially under stress conditions such as cold treatment.

TEs exhibit significant epigenetic differentiation across plant genomes[56]. TE insertions can recruit repressive histone modifications, leading to the formation of new heterochromatin[57–59]. Various epigenetic modifications also play crucial roles in regulating TE transcription and transposition[60,61]. To investigate the epigenetic characteristics of pTEs under cold stress in rice, leaf samples were collected under normal and cold stress (72 h at 4–6 °C) conditions for chromatin immunoprecipitation sequencing (ChIP-seq) targeting H3K27me3 histone modification and for whole-genome bisulfite sequencing (BS-seq). Our analysis revealed distinct distribution patterns of epigenetic modifications across the genome. DNA methylation was relatively uniform across chromosomes, whereas H3K27me3 was enriched, primarily in heterochromatin, at chromosome ends and in centromeric regions (Supplementary Fig. 12). We categorized the genomic sequences into three types, pTE, consensus TE (cTE), and non-TE, for further analysis (Fig. 2d). Under cold stress, the number of H3K27me3 enrichment peaks on cTEs increased slightly, whereas a substantial increase was observed in pTEs, with similar changes observed near genes (Fig. 2e). With respect to methylations, pTEs presented a significant increase in methylation levels following cold stress, whereas changes in cTEs were negligible (Fig. 2e). Additionally, we observed that specific histone modifications, particularly H3K9me2, were generally more enriched in pTEs than in cTEs, marking approximately 27% of the cTEs compared with 46% of the pTEs (Fig. 2f). We also observed notable differences between *indica* and *japonica* with regards to H3K9me2 modifications in genes and their surrounding sequences (Fig. 2g), suggesting that these regions may act as regulatory sequences. These results indicate that pTEs may act as potential transcriptional regulatory regions in the rice response to cold stress.

## Co-expression networks of TEs and genes in response to cold stress

TEs can be expressed specifically under stress conditions and participate in the regulation of gene expression[62,63]. Therefore, we analysed the genome-wide relationship between gene expression and TE

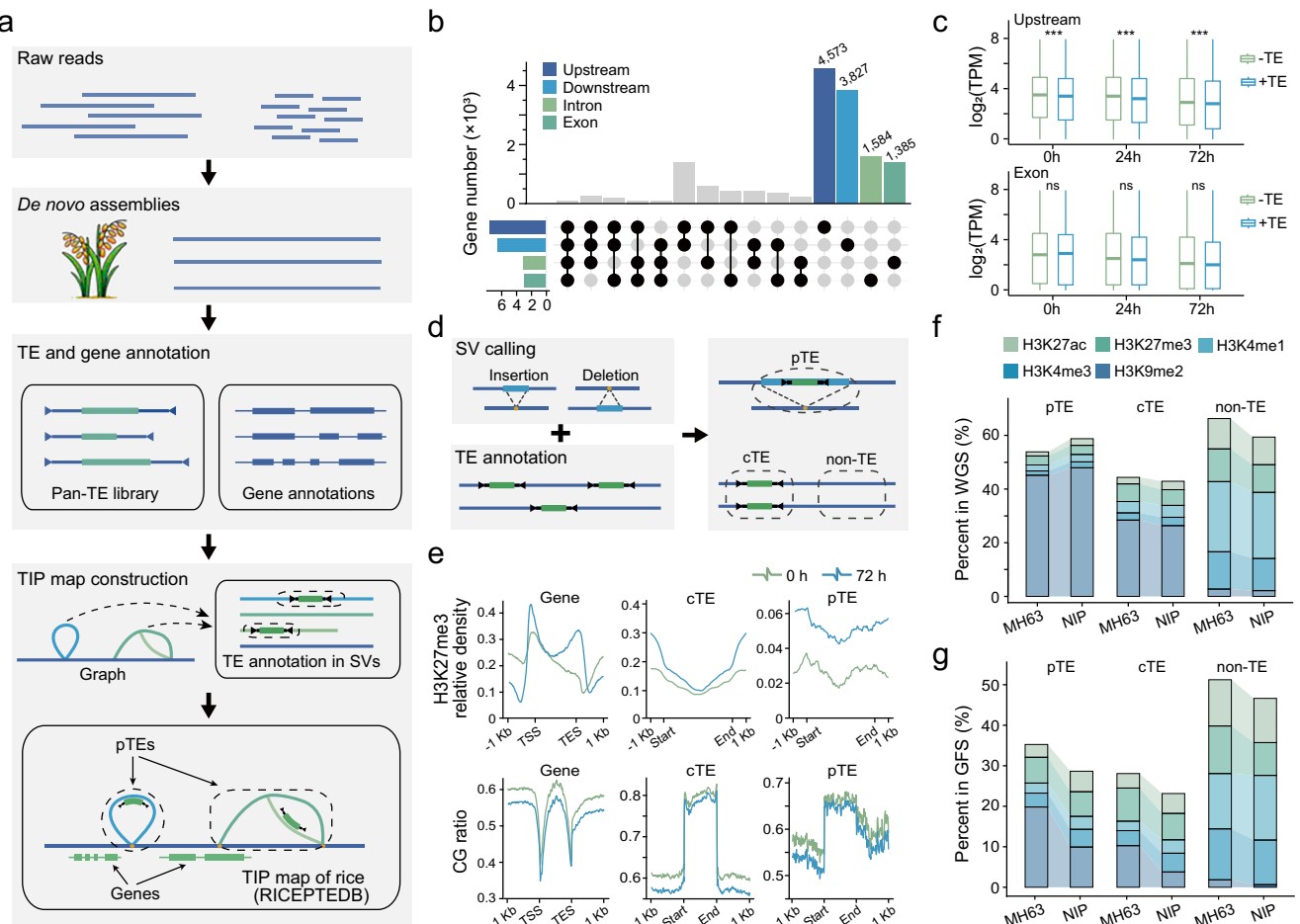

**Fig. 2 | Transcriptional and epigenetic features of pTEs. a** The four-step pipeline we used to construct a rice TIP map. First, the rice genome was sequenced to obtain raw reads, including both long reads and short reads. Subsequently, de novo assembly was performed, followed by the annotation of TEs and genes. Finally, a pangenome graph was constructed, from which SVs were extracted for TE annotation. This information was then integrated with gene annotations to generate the rice TIP map. The SVs annotated as TEs are referred to as pTEs. **b** Bar plot illustrating the number of genes at various genic regions with pTE insertions. The numbers above the bars represent the number of genes with a single pTE insertion. The gray bars represent the gene number with pTE insertions at multiple locations. **c** The impact of pTE insertions on gene expression levels at different cold treatment time points. pTEs in different rice varieties are categorized into two groups: those with TE ( + TE) and those without TE (-TE). "Upstream" indicates that a single pTE is located within 2 kb of the gene and inserted in the upstream region, while "Exon"

signifies that a single pTE is also located within 2 kb of the gene but inserted within the exon. Box plots show the distribution of gene expression levels: the center line represents the median, the box bounds indicate the 25 and 75th percentiles, the whiskers extend to the minimum and maximum values. For the "Upstream" category, $n = 4573$ genes; for the "Exon" category, $n = 1385$ genes. Statistical analysis of these data was performed using a two-tailed Wilcoxon test (***$P < 0.001$, ns: $P > 0.05$). **d** Workflow for classifying genomic sequences into pTE, consensus TE (cTE), and non-TE categories. The two genomes used for comparison are Nipponbare[97] and MH63[38]. **e** H3K27me3 enrichment and epigenetic profiles surrounding genes, cTEs and pTEs. **f** Proportions of different histone modifications among whole genome sequences (WGS) of different categories. **g** Proportions of different histone modifications among gene flanking sequences (GFS) of different categories. GFS refers to the sequences of genes and their flanking 2 kb within the genome. Source data are provided as a Source Data file.

expression under cold stress using transcriptome datasets from different cold treatment periods. PCA of these samples revealed significant differences in expression levels between cold-sensitive and cold-tolerant varieties across different time points (Supplementary Fig. 13). Compared with the 0 h time point, we identified 24,478 cold-responsive genes and 26,914 cold-responsive TEs by combining data from both the 24 h and 72 h cold treatment periods (Supplementary Fig. 14). As the duration of cold treatment increased, the number of cold-responsive genes and TEs also increased. More genes were downregulated than upregulated, whereas the opposite pattern was observed for TEs, where upregulated elements outnumbered downregulated ones (Supplementary Fig. 14). These findings suggest that under cold stress, gene expression in rice is generally suppressed, whereas TE expression is activated, which is consistent with the findings of a previous study[64].

To infer the potential biological functions of TEs, we constructed a co-expression network that links TEs with coding genes on the basis of the correlation between their expression levels. The network included 11,191 genes and 19,566 TEs, with 4,374,619 edges: 3,513,271 connecting gene pairs, 36,712 connecting TE pairs, and 824,635 connecting coding genes to TEs. Through chromosomal location analysis, we identified 418 genes within 100 kb of 603 TE loci, indicating potential cis-regulation of these genes by nearby TEs. However, a relatively large proportion of the identified TEs appeared to participate in trans-regulation (Fig. 3a). We then conducted Gene Ontology (GO) enrichment analysis on the genes co-expressed with cis-TEs and trans-TEs, and found that the functions of these genes are strongly associated with the response of an organism to abiotic stress. Furthermore, genes co-expressed with both cis-TEs and trans-TEs were significantly related to the noncoding RNA metabolic process (Fig. 3b,

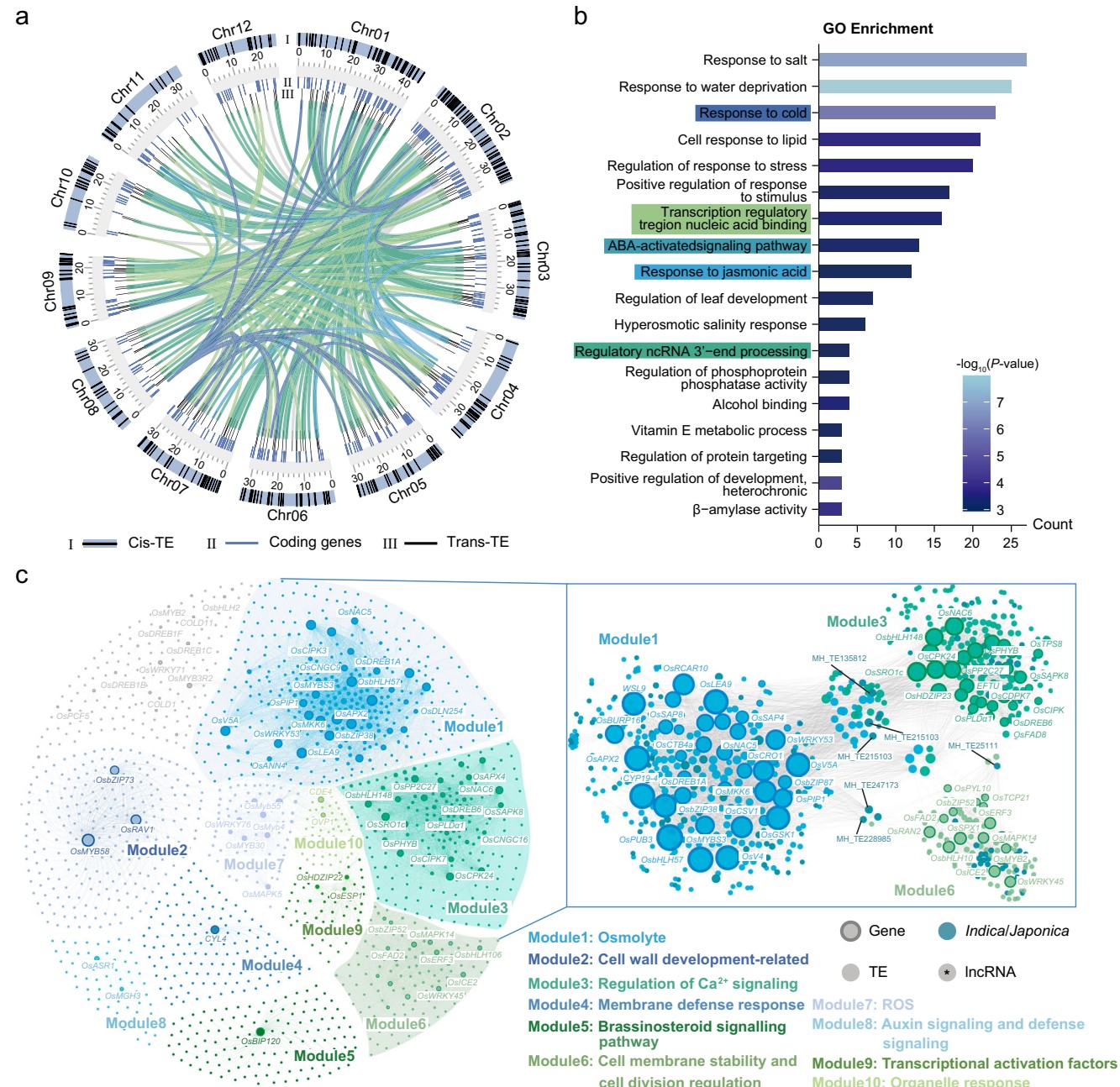

**Fig. 3 | Cold-responsive TEs and genes identified by transcriptome analysis.** **a** Circos plot showing the loci of cis-TEs, trans-TEs, and co-expression coding genes. The internal links represent the correspondence between cold-responsive co-expression genes and their associated trans-TEs. The numbers indicate the length (Mb) of each chromosome. **b** GO enrichment analysis of coding genes associated with trans-TEs. GO terms are ranked by the number of enriched genes, with colored terms indicating associations with cold response, corresponding to the colors in Fig. 3a. GO enrichment analysis was performed using a one-tailed (right-tailed) Hypergeometric test, with *P* values adjusted using the Benjamini-Hochberg method to control the false discovery rate. **c** Co-expression network of known cold tolerance genes and cold-responsive TEs. Different module colors represent gene sets with similar functions. The right subnetwork illustrates the co-expression relationships between genes and TEs in modules 1, 3, and 6. Genes, TEs, conserved TEs between *indica* and *japonica*, and TEs expressing long noncoding RNAs (lncRNAs) are marked with corresponding symbols. The labeled TEs represent cold-responsive TEs with high connectivity and potential functions. Source data are provided as a Source Data file.

Supplementary Fig. 15 and Supplementary Data 3). Consistent with a previous study that TEs influence host gene networks through long noncoding RNAs (lncRNAs) or TE-derived transcriptional regulators[65], we identified 337 TEs associated with the production of lncRNAs in cis-TEs and 10,755 in trans-TEs, which may have an impact on rice cold tolerance.

To further explore the relationships between TEs and cold-responsive genes, we curated previously reported cold tolerance genes in rice (Supplementary Data 4) and constructed a co-expression subnetwork linking these genes to 1,529 TEs. These genes and TEs were organized into 10 modules on the basis of gene annotation (Fig. 3c). Within this co-expression subnetwork, 762 TEs were identified as mediators of TE-derived lncRNAs. Upon examining the positional relationships between these TE-derived lncRNAs and genes, we found that 4 TEs acted as both cis-TEs and trans-TEs, whereas the remainder were classified only as trans-TEs. To investigate potential interactions

between cold tolerance genes and TE-derived lncRNAs in this co-expression subnetwork, we utilized LncTar[66] to predict RNA targets of the lncRNAs. The analysis revealed interactions between 250 TE-derived lncRNAs and 112 cold tolerance genes (Supplementary Data 5). Furthermore, integrating the TIP map revealed that 55 of these 250 TE-derived lncRNAs differentiated between *indica* and *japonica* (Supplementary Data 5).

Finally, we conducted an association analysis between pTEs and the expression levels of genes across different time points, which identified expression quantitative trait loci (eQTLs). These eQTLs may harbor pTEs associated with cold responses. For example, the expression of the TFIIIA-type zinc finger protein-encoding gene OsMH_03G0586900 was significantly associated with a 184 bp insertion upstream 1,318 bp of the gene (Supplementary Fig. 16), indicating that genotypes with the insertion presented increased cold tolerance relative to those without it. These findings suggest that TEs are likely transcriptionally activated in response to cold stress and may regulate cold-responsive gene expression through the production of lncRNAs.

### pTEs associated with rice cold tolerance

To investigate the impact of pTEs on cold tolerance in rice, we analysed previously reported cold-responsive genes (Supplementary Data 4) and identified 54 genes containing pTEs (Supplementary Data 6). *OsTMF* negatively regulates cold tolerance in rice by modulating cell wall properties[67]. This gene also contains a conserved pTE in its upstream promoter region (Supplementary Fig. 17). In Nipponbare, we observed downregulation of the expression of this gene after cold treatment for 72 h, along with repressive modifications in the promoter region (Supplementary Fig. 18), suggesting that TE insertion may reduce the expression of this gene in *japonica* rice compared with *indica* rice, thereby contributing to increased cold tolerance in rice.

We utilized the TIP map to identify cold-tolerance genes. We selected 165 rice accessions from the 3k Rice Genomes Project[1] and collected survival rate data after cold treatment from our previous study[10] (Supplementary Data 1). We used published Illumina data from the 3k Rice Genomes Project to perform TIP genotyping (see Methods) and identify multiple candidate loci associated with reported cold tolerance genes through TIP-GWAS (Fig. 4a and Supplementary Table 6). The SNP-GWAS results also revealed nearly identical loci, although their significance was considerably lower than that of the TIP-GWAS results (Fig. 4b, c). We then validated the biological functions of two genes, Os10g0573700 (*OsCACT*) and Os10g0579800 (*OsPTR*). In *japonica*, the 699 bp downstream region of *OsCACT* contains a 90 bp deletion allele (Fig. 4d and Supplementary Table 7), which has been annotated as a MITE/*Stowaway* element. Compared with *indica*, *OsPTR* in *japonica* contains a 7,734 bp allelic insertion (Fig. 4e), annotated with three types of TEs: DNAnona/*Mutator*, DNA/*Helitron*, and DNAauto/*Mutator*. We obtained the haplotypes of these two genes in 165 rice accessions and found significant differences in survival rates among the different haplotypes after cold treatment, with notable *indica*–*japonica* differentiation (Fig. 4f, g).

Different haplotypes of *OsCACT* and *OsPTR* not only present significant differences in survival rates under cold stress but also show substantial variation in gene expression levels (Fig. 5a, b and Supplementary Fig. 19). Specifically, accessions harboring the downstream pTE insertion in *OsCACT* presented significantly reduced expression of this gene, as confirmed by RNA-seq analysis (Fig. 5a and Supplementary Fig. 20a). *OsCACT* expression was upregulated after 24 h of cold treatment, indicating its early response to cold stress. Previous studies have reported that TE insertions near genes can lead to transcriptional repression[68,69], potentially through DNA methylation-mediated mechanisms[70,71]. To examine this, we compared DNA methylation levels at the *OsCACT* locus between *japonica* rice (NIP, without pTE) and *indica* rice (9311, with pTE). Our analysis revealed that the pTE insertion in *indica* 9311 resulted in high levels of DNA methylation (>

80%), whereas the *OsCACT* locus in NIP, which lacks pTE insertion, showed no detectable methylation (~ 0%) (Supplementary Fig. 20b). This pronounced methylation difference correlates with reduced *OsCACT* expression in *indica* rice, supporting the hypothesis that the pTE may modulate *OsCACT* expression via epigenetic regulation. Similarly, the expression of *OsPTR* also significantly differed between the haplotypes under cold stress (Fig. 5b and Supplementary Fig. 21a). Moreover, *OsPTR* expression was upregulated after 72 h of cold treatment, suggesting its involvement in the late phase of the cold response. Further analysis indicated that the pTE insertion is located within an intron of *OsPTR* (Fig. 4e), potentially resulting in the production of alternative isoforms and consequently affecting overall gene expression[31,34]. Consistently, Iso-Seq data revealed transcript isoform differences in *OsPTR* between the two haplotypes that are associated with the presence of this insertion (Supplementary Fig. 21b).

To validate the biological function of *OsPTR*, we assessed the transcriptional response of this gene under cold stress conditions using qRT–PCR and RNA-seq analyses. The expression of the gene significantly increased in samples under cold stress (Fig. 5b and Supplementary Fig. 22), indicating that the gene is a cold-responsive gene. To further investigate the role of *OsPTR* in cold tolerance, we generated loss-of-function *OsPTR* mutants (osptr) and overexpression lines of this gene in the *japonica* rice cultivar Zhonghua11 (ZH11) genetic background[72] (Supplementary Fig. 23). Under normal growth conditions, no significant phenotypic differences were detected between osptr and ZH11 at either the two-leaf stage or the four-leaf stage (Fig. 5d). After 5 days of cold treatment at 4 °C, the osptr cultivar presented less leaf curling than did ZH11, with an average survival rate of 15%, which was significantly lower than the 91% survival rate recorded in ZH11 at the two-leaf stage (Fig. 5d, e). Compared with ZH11, both the osptr and overexpression lines presented significantly different survival rates at the four-leaf stage, with osptr exhibiting cold sensitivity, whereas the overexpression lines presented increased cold tolerance (Fig. 5d, e). During the two-leaf stage, we measured the relative electrolyte leakage rate of the osptr mutants subjected to cold treatment for 1 to 5 days (Fig. 5g). The results indicated that as the duration of cold treatment increased, the damage to plant cell membranes also intensified. Additionally, we assessed the levels of reactive oxygen species (ROS) in osptr mutants after 3 and 6 days of cold treatment using 3, 3′-diaminobenzidine (DAB) and nitro blue tetrazolium (NBT) staining (Fig. 5f). The results demonstrated that the mutants experienced more severe ROS damage than did ZH11 under cold stress. These findings suggest that *OsPTR* enhances cold tolerance in rice seedlings.

Another gene of interest is *OsCACT*. We assessed the transcriptional response of this gene under various stress conditions using qRT–PCR. The expression of the gene significantly increased in samples under cold and drought stress but decreased under salt stress (Fig. 5c), indicating that the gene may play an important role in plant stress resistance. To further investigate the biological function of *OsCACT* in cold tolerance, we successfully constructed knockout transgenic lines using CRISPR-Cas9 and overexpression lines of this gene in the ZH11 background (Supplementary Fig. 24). In the knockout lines, we observed a significant reduction in the abundance of two carnitine metabolites (Supplementary Fig. 25). These transgenic lines, along with ZH11, were subjected to cold treatment at 4 °C for 5 days. Compared with ZH11, the knockout lines presented greater cold sensitivity at the two-leaf and four-leaf stages, whereas the overexpression lines presented greater cold tolerance at the four-leaf stage (Fig. 5d). In terms of survival rates, the knockout lines presented a significantly lower survival rate than did ZH11, whereas the overexpression lines presented complete survival, with a markedly higher survival rate than did ZH11 (Fig. 5e). During the two-leaf stage, we measured the relative electrolyte

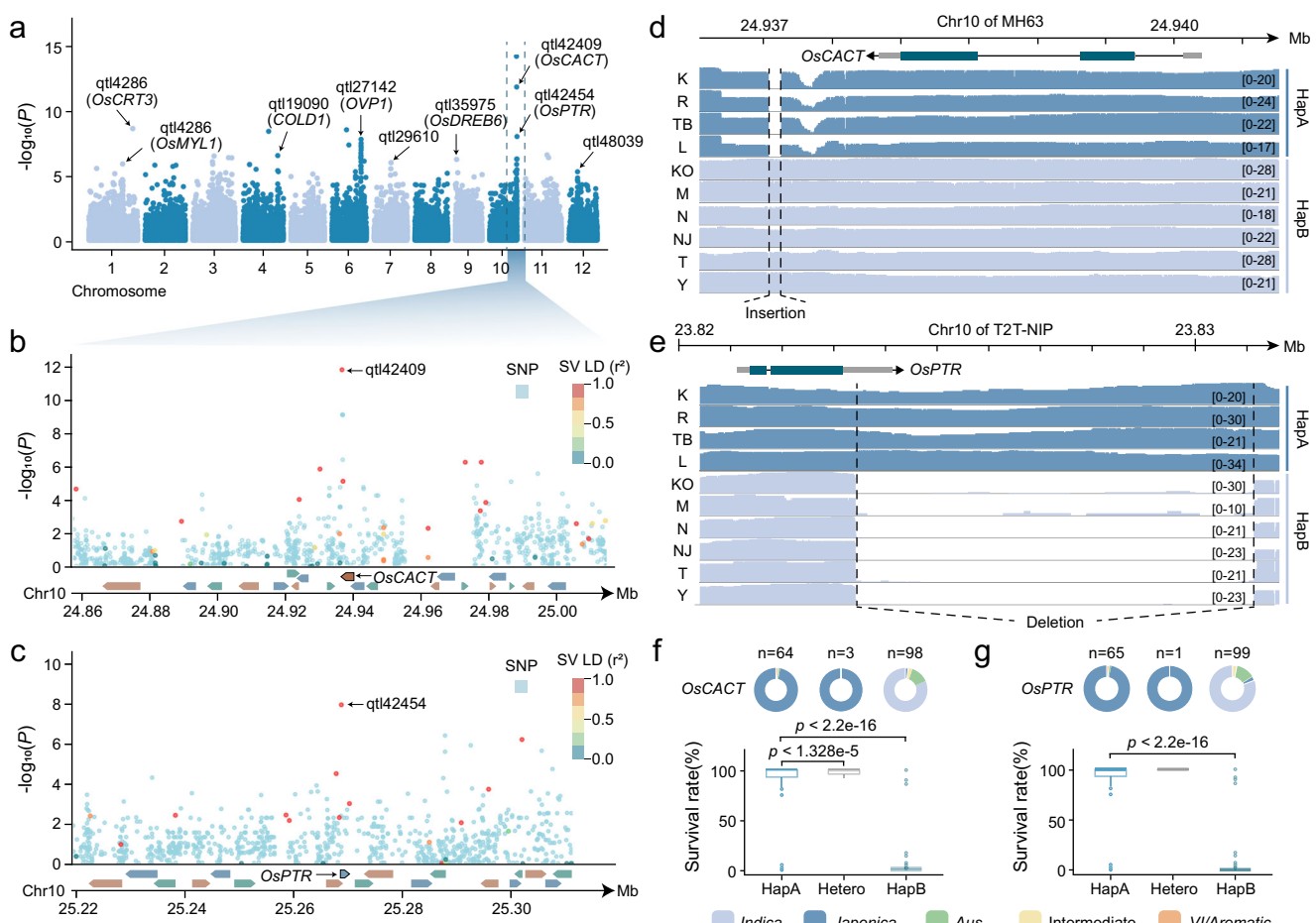

**Fig. 4 | GWAS analysis using the rice TIP map. a** Manhattan plot of GWAS for rice survival rates under cold stress using SVs based on the MH63 genome. The labeled quantitative trait locus (QTL) loci represent locations with high significance and continuous peaks. The genes in parentheses are previously reported cold-tolerance genes significantly associated with these loci. The two highlighted labels indicate loci strongly associated with *OsCACT* and *OsPTR*. **b, c** Manhattan plots of *OsCACT* (**b**) and *OsPTR* (**c**) associated loci. The dots represent the log-transformed *P* values of single-nucleotide polymorphism (SNP) and SV GWAS. The color of the dots reflects the linkage disequilibrium (LD) between the variant and the labeled lead SV. The bottom plots represent the gene structure at each respective locus. **d, e** ONT reads mapping based on MH63 (**d**) and T2T-NIP (**e**). The top plots display the gene structure diagrams. The four tracks labeled K, R, TB, and L represent haplotype A (HapA) for *japonica*, while the six tracks labeled KO, M, N, NJ, T, and Y represent

haplotype B (HapB) for *indica*. The numbers at the right end of each track indicate the reads coverage, and the area between the two dashed lines represents the insertion (**d**) and the deletion (**e**). **f, g** Comparison of survival rates for different haplotypes of *OsCACT* (**f**) and *OsPTR* (**g**) under cold stress. The donut plots illustrate the distribution in different subpopulations and quantity of the 165 accessions across hapA, hapB, and heterozygous (Hetero) categories. The box plots show the survival rates of accessions with different haplotypes. In the box plots, the center line represents the median, the box bounds indicate the 25th and 75th percentiles, and the whiskers extend to the minimum and maximum values. *n* indicates the number of rice accessions in each haplotype group. Statistical analysis of these data was performed using a two-tailed Wilcoxon test. Source data are provided as a Source Data file.

leakage rate of the oscact mutants subjected to cold treatment for 1 to 5 days (Fig. 5g). The results indicated that as the duration of cold treatment increased, the damage to plant cell membranes also intensified. Additionally, we assessed the levels of ROS in oscact mutants after 3 and 6 days of cold treatment using DAB and NBT staining (Fig. 5f). The results demonstrated that the mutants experienced more severe ROS damage than did ZH11 under cold stress. At the four-leaf stage, physiological assessments revealed that the levels of malondialdehyde (MDA), a marker of lipid peroxidation, increased in all the plant lines following cold treatment. However, the MDA content in the knockout lines was significantly greater than that in ZH11, indicating greater oxidative damage. In contrast, the MDA levels in the overexpression lines remained similar to those in ZH11 after 5 days of cold treatment (Fig. 5h), suggesting that oxidative damage was substantially reduced in the overexpression lines. Furthermore, trypan blue staining revealed deeper staining in the knockout line than in ZH11, indicating extensive cell death (Supplementary Fig. 26). Conversely, the

overexpression lines presented much lighter staining, reflecting greater cell viability (Supplementary Fig. 26). Loss of *OsCACT* was lethal in most cells under cold conditions, whereas the overexpression lines retained high viability. In addition, we constructed *OsCACT* overexpression lines in the cold-sensitive *indica* rice cultivar Huanghuazhan (HHZ) background and found that these lines presented improved survival rates under cold treatment (Supplementary Fig. 27), supporting the broad applicability of *OsCACT* in improving cold tolerance in diverse rice cultivars. Previous studies have also suggested that this gene may increase stress tolerance by influencing fatty acid metabolism[73], facilitating the clearance of ROS[74,75] (Fig. 5i). *OsCACT* may also regulate photosynthesis and nitrogen assimilation at the physiological, biochemical, and molecular levels through carnitine, as well as improve plant growth and cold tolerance by enhancing the antioxidant defense system[76] (Fig. 5i). In summary, these findings demonstrate the effectiveness of TIP-GWAS in identifying cold tolerance genes in rice, underscoring the significant role of pTEs in enhancing cold tolerance.

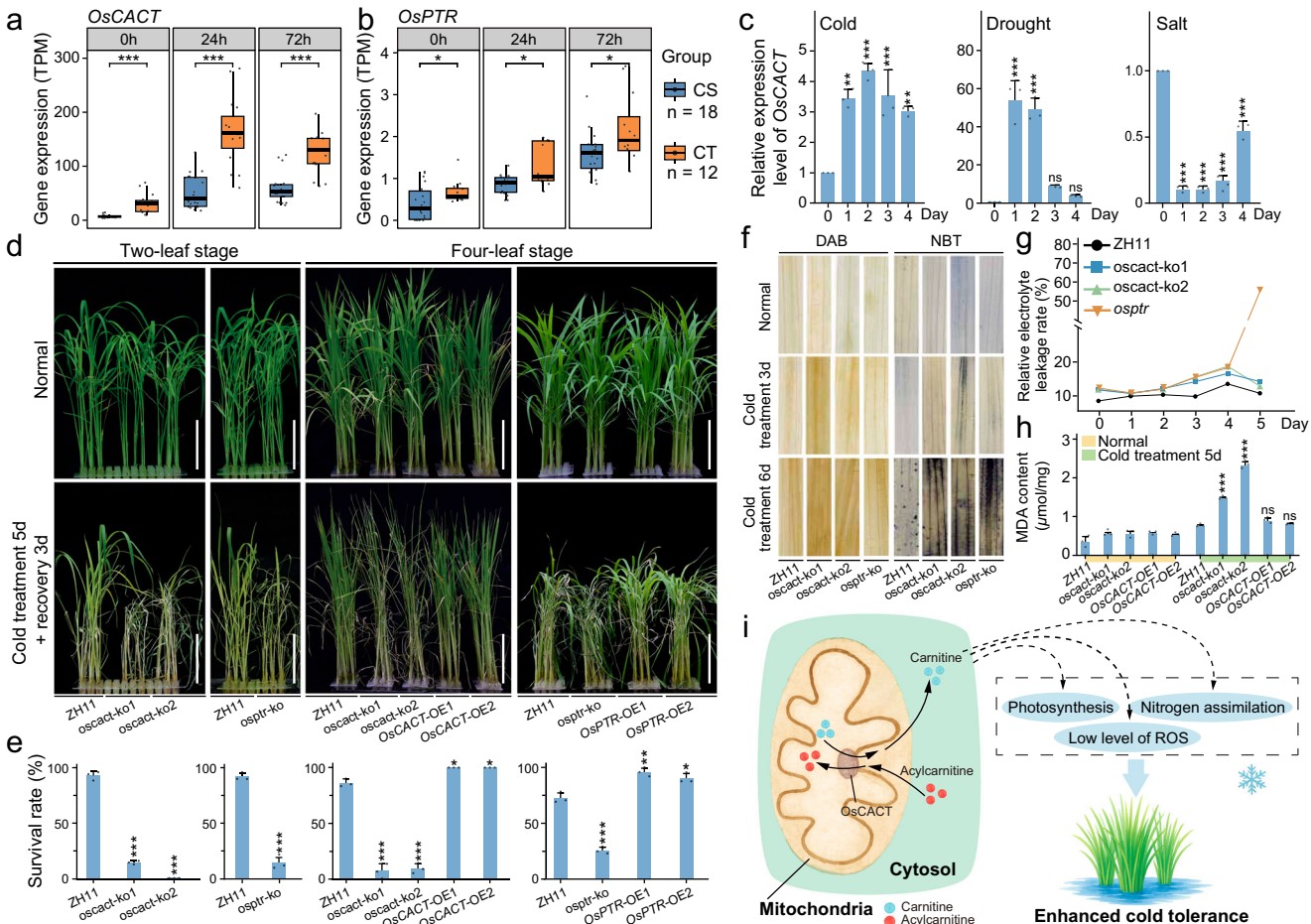

**Fig. 5 | The regulation of cold tolerance in rice by *OsCACT* and *OsPTR*. a, b** Box plots of *OsCACT* (**a**) and *OsPTR* (**b**) expression levels at different cold treatment time points. The different colors represent cold-sensitive (CS) strains, which include six *indica* accessions, and cold-tolerant (CT) varieties, consisting of four *japonica* accessions. In the box plots, the center line represents the median, the box bounds indicate the 25th and 75th percentiles, and the whiskers extend to the minimum and maximum values. For CS (*n* = 6) and CT (*n* = 4), each accession was assessed with 3 biological replicates (total *n* = 18 and 12, respectively). Statistical analysis was performed using a two-tailed Wilcoxon test (*\*P* < 0.05, \*\*\**P* < 0.001). **c** Bar plots of *OsCACT* relative expression levels over time under cold, drought, and salt stress. For each time point, *n* = 3 biological replicates. Statistical analysis was performed using a two-tailed Student's *t*-test, with 0 day as the reference (\*\**P* < 0.01, \*\*\**P* < 0.001, ns: *P* > 0.05). Error bars present mean values ± SD. **d** Representative images of the transgenic lines (*OsCACT* and *OsPTR* knockout mutants and overexpression lines) and their corresponding control of Zhonghua 11 (ZH11) after 5 days of cold treatment followed by 3 days of recovery at the two-leaf and four-leaf stages. Scale bar: 5 cm. **e** Bar plots of plant survival rates after 5 days of cold treatment. For each time point, *n* = 3 biological replicates. Statistical analysis was performed using a two-tailed Student's *t*-test, with ZH11 as the reference (\**P* < 0.05, \*\*\**P* < 0.001). Error bars present mean values ± SD. **f** DAB and NBT staining images of leaves from ZH11 and two genes knockout mutants at different cold treatment time points at the two-leaf stage. **g** Line plots of relative electrolyte leakage rates of two genes knockout mutants and ZH11 over during cold treatment at the two-leaf stage. **h** Bar plots of MDA content in ZH11, *OsCACT* knockout mutants, and *OsCACT* overexpression lines after 5 days of cold treatment at the four-leaf stage. For each time point, *n* = 3 biological replicates. Statistical analysis was performed using a two-tailed Student's *t*-test, with ZH11 as the reference (\*\*\**P* < 0.001, ns: *P* > 0.05). Error bars present mean values ± SD. **i** Model diagram of the mechanism by which *OsCACT* enhances cold tolerance in rice. The expression of *OsCACT* in the mitochondrial inner membrane affects the carnitine content in cells, which in turn influences rice cold tolerance through three distinct pathways. oscact-ko1 *OsCACT* knockout mutant-1, oscact-ko2 *OsCACT* knockout mutant-2; *OsCACT*-OE-1 *OsCACT* overexpression line-1; *OsCACT*-OE-2 *OsCACT* overexpression line-2, osptr-ko *OsPTR* knockout mutant, *OsPTR*-OE-1 *OsPTR* overexpression line-1, *OsPTR*-OE-2 *OsPTR* overexpression line-2, DAB, 3, 3'-diaminobenzidine; NBT nitro blue tetrazolium, MDA malondialdehyde. Source data are provided as a Source Data file.

## Discussion

As a major, globally important crop, rice has been the focus of intense genomics analyses aimed at deciphering its genetic domestication history and informing breeding strategies through multiomics technologies[45,46,48,77]. Recent studies have leveraged multiomics approaches to gain deeper insights into rice domestication[1,78,79]. In this study, we integrated 10 newly assembled rice genomes with the previously published MH63 genome[38] to construct a graph-based rice pangenome. Although the number of de novo assembled genomes used in this study is lower than that used in some other rice pangenome studies[45,46,48], practical evidence suggests that, when target traits are already fixed, the use of an appropriate number of genomes with distinct phenotypic differences is equally effective,

more economical, and suitable for implementation across most research platforms[80–82]. On the basis of the graph, we constructed a TIP map of the rice genome. Our analysis revealed that approximately 18% of the pTEs are highly conserved between the *indica* and *japonica* subspecies, suggesting that these pTEs may have played a role in regulating key genes involved in environmental adaptation and artificial domestication processes following the divergence of these two subspecies. These findings provide valuable resources and tools for further exploration of the role of TEs in rice domestication and environmental adaptation.

The widespread impact of pTEs on rice gene expression is particularly pronounced during the early stages of cold stress (24 h), when the differences in gene expression are more significant. TE

insertions not only cause changes in transcription factor-binding sites, leading to differential gene expression[83] but also significantly influence epigenetic modifications across the genome. Approximately 46% of the sequences of pTEs across the genome are marked by H3K9me2 histone modifications. For pTEs located near genes, H3K9me2 modifications are significantly different between *indica* and *japonica*, whereas other types of histone modifications are not significantly different. These findings highlight the important epigenetic role of TEs in the response of rice to environmental stress and provide further insights into how TEs influence gene expression through epigenetic modifications.

By employing innovative association strategies, TIP-GWAS analysis identified two cold tolerance genes, *OsCACT* and *OsPTR*, both of which were functionally validated using CRISPR/Cas9 technology. Knockout and overexpression experiments on *OsCACT* revealed its critical role in regulating fatty acid metabolism and oxidative stress pathways, which are key mechanisms in enhancing cold tolerance (Fig. 5i). These results not only provide valuable targets for rice breeding programs aimed at improving cold tolerance but also underscore the potential of using pTEs as molecular markers in rice breeding strategies[34,36,84,85].

## Methods

### Plant materials and cold stress treatment
We selected 10 cultivars of Asian cultivated rice (*Oryza sativa*), representing both *indica* and *japonica* subspecies (Supplementary Table 1). The plants were cultivated in an illuminating incubator (RZX-800B, Ningbo Jiangnan Instrument Factory, Zhejiang, China) at the Guangxi University experimental facility in Nanning. The incubator maintained a 13-hour light/11-hour dark cycle, with daytime and nighttime temperatures setting to 30 °C and 24 °C, respectively. Throughout the cultivation period, relative humidity was kept constant at 70%, and light intensity within the incubator was maintained at 12,000 Lux using a three-color, 6000 K energy-saving lamp (Yaming Lighting, Shanghai, China).

Mutant lines for *OsPTR* and *OsCACT* were acquired from the Biogle Genome Editing Center[86], where these mutants were generated using CRISPR/Cas9 technology within the ZH11 and HHZ genetic background. Upon receiving the seeds, primers were designed, and positive plants were identified through PCR and sequencing. These verified lines were subsequently used in the experiments. To examine chilling tolerance of selected rice cultivars, 30-day-old seedlings were transferred to a growth chamber under 4 to 6 °C (dark/light) supplemented with 12,000 Lux of artificial light (GC400/230NG, Gavita, China) and maintained at 70% humidity for 5 days. After 5 days cold treatment followed by 3 days recovery period, phenotypic data were recorded and survival rates calculated. Each treatment group had three independent biological replicates to ensure robustness in the results.

### Genome sequencing, assembly and evaluation
Ten rice accessions were sequenced by the Nanopore and Illumina platforms. Additional data for assembly were obtained from public data[46]. The Nanopore reads were de novo assembled into contigs using NextDenovo (v2.5.0)[87] with default parameters, followed by correction of the assembled contigs with Illumina reads using default parameters in NextPolish (v1.4.1)[88]. Subsequently, these contigs were aligned to the reference genomes MH63[38] for the *indica* subspecies and T2T-NIP[39] for the *japonica* subspecies and manually corrected for chromosome assembly errors using RagTag (v2.1.0)[89]. Genome completeness was assessed using the BUSCO (v5.4.2)[43] embryophyta_odb10 database with default parameters. To further evaluate genome completeness, the genome LTR integrity was calculated using LTR_retriever (v2.9.4)[90] with default parameters. Finally, genome accuracy was assessed using Merqury (v1.3)[40], also employing default parameters.

### Centromere and telomere sequence identification
We utilized CentIER (v2.0)[91] with default parameters to predict the approximate centromere regions of each chromosome based on tandem repeat sequences, retrotransposons, and k-mer frequency distribution. Subsequently, we obtained plant telomere sequences (TTTAGGG/CCCTAAA) from the Telomere Database (https://telomerase.asu.edu/sequences-telomere) and employed Tidk (v0.2.31)[92] with default parameters to predict the telomere regions.

### Collinearity analysis and identification of chromosomal rearrangement events
We conducted pairwise genome alignments between each of the 11 genomes (T2T Nipponbare[39] and our 10 assemblies) and the MH63 genome[38] using the nucmer program in MUMmer (v4.0)[93] with the parameters "-g 1000 -c 90 -l 40". The resulting alignments were visualized to depict the syntenic regions between MH63 and the 11 genomes using GenomeSyn (v1.2)[94] with default parameters. Additionally, the alignments were filtered to retain only those alignment blocks with the parameters "-i 90 -L 100". Chromosomal rearrangement events were identified using SyRI (v1.6.3)[95] with default parameters.

### Transposable elements annotation
We performed TEs annotation on 11 rice genomes (MH63 and our 10 assemblies). First, each genome was individually annotated for TE structure and homology using the panEDTA pipeline of EDTA (v2.1)[44] to identify structurally intact TEs. Subsequently, the panEDTA algorithm was employed to identify and retain sample sequences within individual genomes that contained at least three full-length copies. Redundant sequences in the pan-TE library were then removed using the 80–80 rule[27], resulting in a library of non-redundant TE family sequences, each representing a single sample. This filtered, non-redundant TE library was used to re-annotate all genomes in the pan-genome, allowing for the identification of the total TE content across the 11 genomes, including both structurally intact and fragmented TEs, while maintaining consistent family IDs among the genomes. We ultimately extracted DNA transposons, MITEs, and LTR-RTs to calculate. Structurally intact TEs that could not be classified into known families were named according to their genomic coordinates and considered rare intact TEs. For regions with overlapping TE annotations, our filtering retention principles prioritized the following: when overlapping TEs belonged to the same superfamily, we retained the TE with the higher score; in cases of equal scores, we retained the TE with the longer length. When overlapping TEs belonged to different superfamilies, we considered known nested insertions (LTR into *Helitron*, *Helitron* into LTR; terminal inverted repeat (TIR) into LTR, LTR into TIR) and retained copies that were fully nested within another copy while removing insertions that overlapped with the boundaries of other copies. In instances of overlapping TE annotation boundaries, we retained TEs with stronger structural integrity, prioritizing LTRs first, followed by TIRs, then SINEs and LINEs, *Helitrons*, and finally other TE categories[38,96].

### Gene annotation and pan-gene analysis
We obtained gene annotations of Nipponbare[97,98] and MH63[38] and performed homology annotation of the 10 assemblies using GMAP (v2021.12.17)[99] with default parameters. Following this, we extracted the protein sequences from the genomes using TBtools-II (v2.001)[100] and conducted gene family analysis with BPGA (v1.3)[101].

### Construction of the TIP map
First, we constructed a pan-genome graph by aligning MH63 and the 10 assemblies using minigraph (v0.19)[49] with default parameters. We then converted the GFA format graph to VG format[102] using vg view,

followed by the extraction of SVs from the graph using vg deconstruct. Finally, we aligned the assemblies to the graph using minigraph (v0.19)[49] with default parameters for SV genotyping. In the second step, we merged the TE sequences from the 10 assemblies into a TE library using a Perl script make_panTElib.pl[44]. Following the 80-80 rule[27], we performed blastn to align the SV sequences from the first step to the TE library and further annotated the results using EDTA (v2.1)[44]. We then combined the results of these two steps into a TIP map using a Python script.

## Population structure and phylogenetic analysis

We selected 165 samples from the the 3k Rice Genomes Project and downloaded resequencing data from public databases[1]. Phenotype data were obtained from our previous study[10]. For the GFA file of the pangenome graph, we first constructed an index using VG (v1.55.0)[102] with the parameters "autoindex -w giraffe". Subsequently, we aligned all Illumina data to the graph pan-genome using vg giraffe[103] with default parameters, resulting in GAM files. We then utilized vg pack and vg call with the parameters "pack -Q 5 -s 5" and "call -a" to obtain the final VCF format genotyping results. We downloaded SNP variant data from the 3k Rice Genomes Project[1] and filtered it for the 165 samples using VCFtools (v0.1.16)[104]. PCA was performed using PLINK (v2.2.9)[105] with the parameters "--pca 2 header tabs var-wts --out snp.filter --allow-extra-chr --double-id." We filtered the VCF file with PLINK using the parameters "--hwe 0.0001 --make-bed --out snp --double-id," and calculated values of K ranging from 2 to 7 using ADMIXTURE (v1.3.0)[106] with default parameters. We filtered out subspecies with fewer samples and used 164 accessions for visualization. The analysis for TIP followed the same SNP analysis procedures and parameters.

## Transcriptome data processing

We obtained RNA-seq data for 90 samples, consisting of three biological replicates for each of the 10 materials across three time points, from our previous study[10]. We additionally isolated total RNA from 12 samples, including 2 time points for each of the 6 samples from 165 rice accessions (Supplementary Data 2). The total RNA of the samples was extracted using TRIzol, and mRNA was purified from the total RNA using poly-T oligo-attached magnetic beads. According to the manufacturer's instructions, cDNA libraries from rice leaf tissue samples were prepared using the TruSeq Stranded mRNA Sample Prep Kit and subjected to 150-bp paired-end sequencing using HiSeq2500. After filtering the raw reads, the clean RNA-seq data were aligned to the MH63 reference genome using HISAT2 (v2.2.1)[107]. Based on the RNA-seq mapping results, we sorted the BAM files and the generated index files using SAMtools (v1.21)[108] with default parameters. Transcripts were assembled using StringTie (v2.2.0)[109] with default parameters. Gene expression was measured by StringTie as fragments per kilobase of transcript per million fragments mapped (FPKM) and transcripts per kilobase million (TPM). Differential expression genes (DEGs) were identified using DESeq2 (v1.18.1)[110]. To analyze gene and TE expression under the same conditions, we utilized TE transcripts (v2.2.3)[111] with default parameters to compute the expression matrices for both genes and TEs. To evaluate the consistency among biological replicates, we calculated the correlation between biological replicates using Deeptools (v3.5.6)[112] and performed PCA on the expression levels of all detected genes and TEs. Using gene expression levels under normal conditions (0 h) as a reference, we identified differentially expressed genes at 24 h and 72 h based on the criteria of $|\log_2(\text{Fold Change})| \geq 1$ and FDR < 0.05. Genes exhibiting differential expression at any time point and in any cultivar were classified as cold-responsive genes. Similarly, cold-responsive TEs were identified using the same method.

## Processing of ChIP-seq and BS-seq data

We conducted cold treatment and leaf tissue extraction on two rice varieties with differing cold tolerance: 9311 (cold-sensitive) and Nipponbare (cold-tolerant). Leaf samples were collected from both the two materials under normal (0 h) and cold stress (72 h) conditions. ChIP-seq and whole-genome BS-seq were carried out by Shanghai Jiayin Biotechnology Ltd. ChIP-seq was performed using an antibody specific to the histone mark H3K27me3, and the DNA methylation modifications in each sample were detected using BatMeth2[113] with default parameters.

We obtained histone modification data for MH63 and Nipponbare from a public database[114]. Using Burrows-Wheeler Aligner (BWA, v2.2.1)[115] with default parameters, we aligned the raw clean reads to the corresponding genomes. The SAM files were then converted to BAM format, sorted, and indexed using SAMtools (v1.21)[108] with default parameters. Peak calling was performed using MACS2 (v2.2.9.1)[116] with on the basis of a threshold of $P < 1 \times 10^{-10}$.

Based on the chromosomal synteny and SVs identified above between the MH63 and Nipponbare, we categorized the whole-genome sequences into three types: non-TE, pTE, and cTE, using TE annotations from both genomes. We then used BEDTools (v2.30.0)[117] to quantify the histone modifications on each sequence type. Additionally, we extracted the genes from MH63 and Nipponbare along with their flanking 2 kb sequences, classifying them into non-TE, pTE, and cTE types, and similarly quantified histone modifications for each sequence type using BEDTools.

## Iso-seq data processing and identification of TE-derived lncRNAs

We obtained full-length transcripts from our previous study[10]. The corrected high-quality full-length transcripts were aligned to the MH63 genome using minimap2 (v2.25-r1173)[118] with parameters "-ax splice -uf --secondary=no -G 10,000". Full-length and non-redundant transcripts from 30 sequencing samples were merged using Transcriptome Annotation by Modular Algorithms (TAMA)[119] to generate a unique set of full-length transcripts. Comprehensive transcript characterization was then performed using SQANTI3 (v4.2)[120], which employed a machine learning algorithm to filter out false-positive transcripts.

To identify lncRNAs, transcripts with lengths ≥ 200 nt and open reading frames (ORFs) ≤ 120 amino acids were first screened using ORFinder. To remove transcripts potentially encoding short peptides, blastx was run with the parameters "-e 1.0e-4 -S 1" against the Swiss-Prot database. Overlaps with rRNA, tRNA, sRNA, and miRNA entries in the Rfam database were also filtered out. Coding potential of the remaining transcripts was evaluated using CPC2[121], PLEK[122], PLncPRO[123,124], and RNAplonc[123,124]. Only transcripts identified as non-coding by at least two of these tools were classified as lncRNAs. Finally, to identify TE-derived lncRNAs, the transcripts classified as lncRNAs were overlapped with MH63 TE annotations to produce a set of TE-derived lncRNAs.

## Construction of a co-expression network

We performed a co-expression analysis of gene and TE expression matrices from 90 rice samples collected before and after cold stress. The matrices for TE and gene expression levels have been previously described in the "Transcriptome data processing" section. Four tools were used to construct gene and TE co-expression networks: WGCNA[125], Pearson correlation coefficient (PCC)[126–128], CORNET[129], and TEffectR[130]. The parameters used in each tool were as follows: (1) In WGCNA, the blockwiseModules function was employed to construct unsigned networks with the following parameters: power=5; TOMType=unsigned; reassignThreshold=0; mergeCutHeight=0.25; numericLabels=TRUE; pamRespectsDendro=FALSE. (2) The quantification results of genes and TEs, generated by TEtranscripts, were used to build a linear regression model (LM) with the TEffectR package. The covariates were set to the three time points of cold treatment (normal, 24 h, 72 h), and an adjusted $R^2 \geq 0.9$ was required. (3) The average FPKM values from replicate experiments for genes and TEs were log2-transformed after adding a pseudocount of "1". Quantile normalization

was performed using the preprocessCore package in R. The mean expression across all experiments was subtracted for each gene or TE to eliminate potential batch effects. We then calculated PCC using the cor function in R, with co-expression considered significant when the absolute value of PCC was ≥ 0.55[128,131,132]. (4) The online tool CORNET was also used to analyze the co-expression of gene and TE expression matrices, with default parameters.

To ensure the robustness of the results, co-expression relationships were only retained if detected by at least two software tools. The biological function of TEs was predicted based on their positional relationship with protein-coding genes (cis-TE) and expression correlation (trans-TE). TEs located within 100 kb upstream or downstream of coding genes were classified as cis-TEs, while trans-TEs were identified through expression correlation analysis. We visualized the subnetwork of 518 cis-TEs co-expressed with genes and the top 6000 trans-TEs co-expressed with genes using Circos[133]. Subsequently, the subnetwork of 138 known cold-tolerance genes and their co-expressed TEs was visualized using Gephi (v0.10.1)[134].

## LncRNA-mRNA interaction prediction

To further examine the interactions between TE-derived lncRNAs and target genes in the co-expression network, we used LncTar[66]. The fasta sequences of lncRNAs and gene transcripts were input into LncTar, with the threshold normalized deltaG (ndG) set to -0.05. Transcripts with calculated ndG values below this threshold were identified as targets of the respective lncRNAs, with the corresponding genes designated as target genes of the lncRNAs.

## SNP calling and GWAS analysis

Clean Illumina reads downloaded from previous studies[1] were mapped to MH63 genome using BWA (v2.2.1)[115] with default parameters. Genomic variants were identified with HaplotypeCaller and the GenotypeGVCFs functions in the Genome Analysis Toolkit (GATK, v4.2.6.1)[135]. Then BCFtools (v1.19)[136] was used for vcf files merging with default parameters. SNP calls were further filtered based on the parameters "--maf 0.05 --max-missing 0.9".

The phenotype under investigation was the survival rate of 165 rice seedlings subjected to 5 days of cold treatment followed by 3 days of recovery. These 165 rice cultivars represent the major rice subpopulations globally, encompassing 14 *aus*, 23 *indica* I, 9 *indica* II, 27 *indica* III, 21 *indica* intermediate, 3 intermediate, 12 *japonica* intermediate, 27 temperate *japonica*, 28 tropical *japonica*, and 1 VI/*aromatic*. To evaluate the cold tolerance of these selected rice cultivars, 30-day-old seedlings were placed in a growth chamber maintained at 6 to 8 °C (dark/light) with 12,000 Lux of artificial light (GC400/230NG, Gavita, China) and 70% relative humidity for 5 days. Following this, the plants were transferred to room temperature for 3 days, and their phenotypic responses were documented, as detailed in Supplementary Data 1.

We merged all VCF files generated from SV genotyping of Illumina data using BCFtools. First, we indexed the VCF files, followed by merging these files with the parameters "-0 -l vcf_gz_file.list -L 2 -m id -O v". The merged VCF file was then converted with PLINK (v2.2.9)[105] for subsequent analysis using default settings. The VCF file was filtered with the parameters "--recode 12 --output-missing-genotype 0 –transpose". To investigate the relationship between SVs and phenotype (Supplementary Data 1), we performed GWAS using the mixed linear model in EMMAX[137] with the parameters "-v -d 10", and significant TIP loci were filtered based on SV ID. Genes near pTEs were identified from the RAP database[97,98] for further analysis.

## RNA extraction and qRT–PCR

Total RNA was extracted from rice leaves using the Total RNA Extraction Kit (Axygen AxyPrep, USA). The quality and concentration of the RNA samples were assessed using a microplate spectrophotometer (Epoch 2, BioTek, USA). First-strand cDNA was synthesized from 500 ng of RNA for each sample using HiScript III RT SuperMix for qPCR (+ gDNA wiper) (Vazyme, China). The qPCR reaction mixtures were prepared using ChamQ Universal SYBR qPCR Master Mix (Vazyme, China). The primer sequences are provided in Supplementary Table 8. qRT–PCR was conducted on a fluorescent quantitative PCR instrument (qTOWER3, Analytik Jena, Germany). The relative expression levels of each gene were evaluated using the $2^{-\Delta\Delta CT}$ method[138], with *Actin* serving as the reference gene.

## Relative electrolyte leakage measurement

The relative electrolyte leakage (REL) from the rice leaves was measured using a conductivity meter (AZ86031, AZ Instrument, China). First, the conductivity value of ddH₂O (S0) was established as a blank control. Rice leaves were cut into 5 mm × 5 mm pieces, with 20 pieces from each group placed in an Eppendorf tube containing 20 mL of ddH₂O. The tubes were rotated at a constant speed for 1 hour, after which the initial conductivity value (S1) was recorded. The Eppendorf tubes were then incubated in a boiling water bath for 10 min, and the final conductivity value (S2) was measured after a subsequent 10-minute equilibration at room temperature. The REL was calculated using the following formula[139]:

$$REL(\%) = \frac{S1 - S0}{S2 - S0} \times 100\% \tag{1}$$

## DAB, NBT and trypan blue staining and quantification of malondialdehyde (MDA) contents

H₂O₂ and O₂⁻ in the leaf were detected using DAB and NBT[140]. In brief, two-leaf stage seedlings were immersed in freshly prepared 1 mg/mL DAB staining solution (pH 3.8) or in a 50 mM sodium phosphate buffer (pH 7.0) containing 0.05% NBT and 10 mM NaN₃. The seedlings were then vacuum-infiltrated at 0.8 psi for 30 min. After infiltration, they were incubated in the dark at 25 °C with gentle shaking at 100 rpm for 6 h. Subsequently, the seedlings were bleached using a solution of acetic acid:glycerol:ethanol (1:1:3, v/v/v) at 65 °C for 2 h and stored in 95% ethanol until imaging.

Trypan blue staining of ZH11, *OsCACT* knockout mutants, and *OsCACT* overexpression lines was performed on the second leaf at the seeding stage[141]. Briefly, leaves were immersed in trypan blue staining solution (30 mL ethanol, 10 g phenol, 10 mL H₂O, 10 mL glycerol, 10 mL lactic acid, and 10 mg trypan blue), boiled for 2 to 3 min, and then cooled to room temperature for 1 hour. Samples were destained by boiling in 2.5 g/mL chloral hydrate solution for 20 min, with 2 to 3 changes of the solution at room temperature. The stained leaves were stored in 50% (v/v) glycerol until imaging.

Fresh leaf samples (0.5 g) were collected before and after cold treatment and homogenized in 5 mL of 5% (w/v) trichloroacetic acid (TCA) solution. Following centrifugation at 12,000 × *g* for 5 min at 4 °C, 2 mL of the supernatant was mixed with 2 mL of 0.67% (w/v) thiobarbituric acid (TBA). The resulting mixture was incubated in boiling water for 30 min, cooled to room temperature, and then centrifuged at 5000 × g for 10 min. The supernatant was collected for absorbance measurement at wavelengths of 450, 532, and 600 nm. The MDA concentration was calculated according to the following formula:

$$MDA\left(\frac{\mu mol}{mg}\right) = \frac{\left[6.45 \times (OD_{532} - OD_{600}) - 0.56 \times OD_{450}\right] \times V \times A}{W} \tag{2}$$

where $OD_{532}$, $OD_{600}$, and $OD_{450}$ represent the absorbance value at 532, 600, and 450 nm respectively; $V$ (mL) is the total reaction volume; $A$ is the dilution ratio at the final determination of the sample; and $W$ (mg/mL) is the protein content in the supernatant of the homogenate[142]. All experiments were carried out with at least 3 replicates.

## Metabolomic analysis

Samples were collected from both ZH11 and oscact-KO at two time points: normal conditions and 5 d cold treatment for the comparison of metabolite differences before and after cold treatment. Untargeted metabolomics was performed with three biological replicates for each sample. All metabolite extraction, detection, and quantification were carried out by Shanghai Applied Protein Technology Co., Ltd. Metabolite identification was achieved by comparing the accurate *m/z* values (<10 ppm) and MS/MS spectra against an in-house database[143,144] established with authentic standards.

## Reporting summary

Further information on research design is available in the Nature Portfolio Reporting Summary linked to this article.

## Data availability

The raw sequencing data generated in this study have been deposited in the National Genomics Data Center (NGDC) under accession PRJCA032145. The previously released transcriptome data and resequencing data used in this study are available at NGDC under accession PRJCA017960[10]. All genome assemblies, annotations and TIP information are available at Rice pTE Database (https://cbi.gxu.edu.cn/RICEPTEDB/) and Zenodo [https://doi.org/10.5281/zenodo.16625002][146]. Source data are provided with this paper.

## Code availability

Code used for the pangenome analysis is available at Zenodo [https://doi.org/10.5281/zenodo.16625002][146].

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

## Acknowledgements

This study was supported by the National Natural Science Foundation of China (32100526, U24A20369 and 32260446), Guangxi Natural Science Foundation (2024GXNSFGA010003), Special fund for youth team of the Southwest Universities (SWU-XJPY202306), Special Key Project of Technological Innovation and Application Development of Chongqing (CSTB2024TIAD-KPX0063), the Guangxi S&T Program (Guike-AD25069107), the Young Elite Scientists Sponsorship Program by CAST (2022QNRC001), Supported by the earmarked fund for CARS-12, the Fundamental Research Funds for the Central Universities of China (SWU-KR24030 and SWU-KF25037) and Innovation Project of Guangxi Graduate Education (YCBZ2024048).

## Author contributions

J.-M.S., L.-L.C., J.L., and K.L. designed studies and conceived the project. Y.Q., Z.Z., T.O., D.L., and R.Q. carried out the analysis. K.L., T.O., R.L., P.G., Y.T., and M.Q. managed the field work and prepared the samples. J.-M.S., L.-L. C., J.L., K.L., Y.Q., Z.Z., T.O., and D.L. wrote the manuscript. J.-M.S., L.-L.C., J.L., K.L., J.-N.L., and Z.L. revised the manuscript. All authors read and approved the final version of the manuscript.

## Competing interests

The authors declare no competing interests.
