## [Peer Review file · Nature Communications]

Pangenome analysis of transposable element insertion polymorphisms reveals features underlying cold tolerance in rice

Corresponding Author: Professor Jiaming Song

Version 0:

Reviewer comments:

Reviewer #1

(Remarks to the Author)

This paper presents a comprehensive analysis of the effect of pTE on cold resistance in rice. There are many analyses performed on different accessions, which makes reading difficult, and also to weigh each piece of evidence.

At the end of the day, the most convincing is the editing of two genes potentially involved in cold resistance. Yet, even this is not completely convincing. Relative to OsPTR (lines 345-351), is the ZH11 strain natively resistant? After five days, what is the difference in survival? These figures could be in the main ms.

As for OsCACT, evidence between the gene and cold resistance seems more solid. Yet, the relationship between pTE and OsCACT expression is not causally proved, in my opinion.

Overall, authors should be more cautious in their writing and nuance their assertions.

Specific comments

line 27-29: This is a strong conclusion, not really proved.

l 61: any reference for link between TE and SVs?

l 134 ff: are these numbers similar to other reported previously?

l 211: tone down, 'may suggest' ...

l 218: how were differences in transcription across conditions assessed, is there a statistical test?

l 224. again we cannot tell causation from correlation

l 231: 'reduced', again, is there a significant test? there are very few samples and many variables, a scenario prone for false positives.

l 233: 'influence' not proved, -> 'associated with' or similar

l 258: 'serve' not proved

l 267: what significance level / criterion was used to determine that a gene is cold responsive?

l 276: a coexpression network between TE and coding genes, you mean gene expression and TE genotype?

l 275: GO is really a very confusing world, I suggest to remove but leave that voluntary.

l 293-294: how was this done?

l 305 how were these pTE chosen?

l 306. why lncRNA?

l 319: why different genes here and in previous list (l 295)

l 350: survival rate of both genotypes by stage

l 587: are the 90 samples the same set as mentioned that correspond to 10 different genotypes only? Not clear how TE expression is measured

l 603 : why that value (0.55)

l 618: what is nDG?

l 621ff: what GWAS is done? what is the phenotype?

l 672: what is the metabolomic analysis done for?

Minor comments

k symbol for thousand is lower case, they also use K for number of clusters in Admixture.

Reviewer #2

(Remarks to the Author)

Transposable elements (TEs) are ubiquitous genomic components and play important roles in some traits variations by affecting the expression of their nearby genes in plants. Although an increasing number of rice studies focused on identifying TEs in rice populations and constructing pan-TE maps, which have revealed numerous TE characteristics related to rice domestication and agronomic traits, the influence of TEs on cold tolerance in rice remains largely unexplored. In this study, the authors constructed a pangenome graph and identified 30,316 TIP sites, highlighting significant diversity among pTEs. Elaborated on the contribution of pTEs differential gene expression prior to and following cold stress, and identified numerous cold-responsive genes and pTEs. TIP-GWAS on cold tolerance phenotypes in 165 rice accessions led to the discovery of novel cold-tolerance genes, such as *OsCACT* and *OsPTR*, and overexpression experiments and metabolomic data was validated.

Overall, the logic is good, the analyses are clear and data are reliable. Here are some comments:

1. Many errors/tipos in the manuscript, either from the text of or from the figures, need to carefully checked and revised. Here are only several examples.

(1) Some specialized nouns should be written properly. "long interspersed nuclear element (LINE)" in Line 72 should be "long interspersed nuclear elements (LINEs)", *Stow* should be *Stowaway* in Line 339 and Figure 1d. The authors could follow Wicker et al, 2007 to correct the naming system of transposable elements in the manuscript. Authors are kindly requested to carefully review the entire text.

(2) Some specialized nouns should be italicized, such as *Oryza sativa* in Line 433 and P value in Figure 2 and in Figure 5, etc. Authors are kindly requested to carefully review the entire text.

2. Why wasn't the MITEs family included in Fig1b? It would be better to add MITEs in Figure 1b because the MITEs is also a major part of TEs.

3. Line 157. The obtained TIPs, particularly the heterozygous TIPs, should be validated with more evidences, e.g., the PCR experiments or the IGV browser. And their accuracy rate could be estimated.

4. Line 174-175. There is no figure or table that can support "pTE insertion rates were significantly higher in 4 assemblies, with the peak primarily contributing to TEs shared by the 4 japonica assemblies". In general, when we say "significant", it need to be based on statistical indicators, such as a p-value less than 0.05 or 0.01.

5. Line 186-191. The authors declared that "the distribution suggested a preference for pTE insertion in the upstream 2 kb 191 region" because 9,131 events were identified within the 2kb region and only 7,559 occurred within gene bodies. Typically, when comparing the frequency of TEs, it is necessary to use their density rather than the total number to avoid an incorrect comparison due to different sequence lengths.

6. For TIP-eQTL and TIP-GWAS, whether the TIPs absent in the reference genome are used or not? If not, would this introduce a bias towards the reference genome?

7. Line 456-457, "these contigs were aligned to a reference genome". The reference genome should be clearly indicated. Why was it chosen as the reference genome? This is important because different reference genomes can result in variations in genome size and sequence using RagTag.

8. Fig4.f-g. Since japonica is inherently more cold-tolerant than indica, it would be better to provide both the phenotype and the gene expression data to support the functional differences among different haplotypes of the two genes.

9. Line 1187. Please add the accession numbers of the 10 rice assemblies and their annotation (including TE annotation and gene annotation) in NCBI or other public databases in the Data availability section.

Reviewer #3

(Remarks to the Author)

Here, the authors assembled 10 high-quality rice genomes by various sequencing methods and constructing a pangenome graph; Using the pangenome data and resequencing 165 rice accessions, the authors identified 30,316 transposable element insertion polymorphism (TIP) sites; at the same time, they identified TEs responsive to cold stress from transcriptome data; at last using the data TIP-GWAS revealed novel cold-tolerant genes, *OsCACT* and *OsPTR*. The manuscript provided the valuable pTEs data and valuable targets for breeding cold-tolerant rice varieties. However, some other rice pan-genome studies had many de novo assembled genomes include many varieties of the selected 10 rice accessions, such as Koshihikari, Lemont, Nipponbare, 9311 etc. In addition, why did transposable element insertion or deletion of the two novel cold-tolerant gene change their expression?

1. Since the other rice pan-genome studies had many de novo assembled genomes, the authors should construct the rice pangenome graph by integrating these genomes, and construct a bigger pan-TEs.

2. Supplementary Table 1. The cold-tolerance level of the 10 rice accessions is not even, it need add some accessions on cold treatment for RNA-seq analysis.

3. Fig4, TIP-GWAS revealed novel cold-tolerant genes, *OsCACT* and *OsPTR*, It is not clear that the transposable element insertion or deletion induce or reduce the expression of *OsCACT* and *OsPTR*; it need some data to support the results.

4. Because ZH11 (temperate japonica) is cold-tolerant rice variety. Except checking function of *OsCACT* and *OsPTR* in ZH11, adding a cold-intolerant variety is better.

5. Since the two novel cold-tolerant genes have different allele, it is better using rice NILs material test the cold-tolerant function.

Version 1:

Reviewer comments:

Reviewer #1

(Remarks to the Author)

I am satisfied with changes provided in the new version of the manuscript. There is no file in the code submission list. Zenodo page below not found.

(Remarks on code availability)

Unable to review.

Reviewer #2

(Remarks to the Author)

After revision, the manuscript has been greatly improved. However, some wording is a bit non-standard and it would be good if the authors can have a native English speaker read the manuscript. Please revise and then publish

(Remarks on code availability)

Reviewer #3

(Remarks to the Author)

The authors provided a point-by-point response to my comments in the revised paper. The manuscript identified 30,316 pTEs data and integrated transcriptome and epigenetic modification data; If the data are visualized and easy to be used by breeders, it is valuable for breeding cold-tolerant rice varieties. Though the authors release a web tool (<https://cbi.gxu.edu.cn/RICEPTEDB/>); they just provided the visualized data of MH63RS3 genome, not 131 published high-quality rice genomes, transcriptome and epigenetic data. The authors should integrate the data on the web to visualize it. Some comments:

1. The authors claimed that they made the 131 data through the Rice pTE Database; however, I do not find these data in the web.

2. Fig 5b, the OsPTR expression was downregulated after cold treatment, however, Supplementary Figure 22, OsPTR expression was upregulated after 36 and 72 hours of cold treatment. Why?

3. Figure R2b This differential methylation correlates with the observed downregulation of OsCACT expression. The increase in the methylation of the region between the two dashed lines represents the pTE. The TE methylation reduces the OsCACT? The author should provide the information of TE sequence of methylation. The TE is still high methylation if it is inserted at the other locus?

4. Figure 5d and Supplementary Figure 27 b, in ZH11 background, in normal condition, the oscact-ko lines are higher; however, in HHZ background, the OsCACT overexpression lines are more strong. OsCACT also affects the seedling development?

(Remarks on code availability)

Version 2:

Reviewer comments:

Reviewer #1

(Remarks to the Author)

No further comments.

(Remarks on code availability)

Reviewer #3

(Remarks to the Author)

The revised manuscript has been greatly improved. Especially adding more data on the web tool (<https://cbi.gxu.edu.cn/RICEPTEDB/>). It is valuable for breeders and researchers. It meets the standard for publication.

(Remarks on code availability)

Reviewer #1 (Remarks to the Author):

1. *This paper presents a comprehensive analysis of the effect of pTE on cold resistance in rice. There are many analyses performed on different accessions, which makes reading difficult, and also to weigh each piece of evidence.*

Response: Thank you for your valuable feedback. To improve the readability of the manuscript and help readers better understand the context, we have added clarifications about the two major rice subspecies, *indica* and *japonica*, when mentioning MH63, NIP, and ZH11 in line 87, 127, and 383. These additions aim to make the manuscript more accessible and facilitate comprehension of the analyses presented.

2. *At the end of the day, the most convincing is the editing of two genes potentially involved in cold resistance. Yet, even this is not completely convincing. Relative to *OsPTR* (lines 345-351), is the ZH11 strain natively resistant? After five days, what is the difference in survival? These figures could be in the main ms.*

Response: Thank you for pointing out these important issues. First, we confirm that the *japonica* rice variety ZH11 is a naturally cold-resistant cultivar¹, as demonstrated in our manuscript. In response to your suggestion, we have moved all figures related to *OsPTR* from the supplementary materials to the main figures (**Figure 5**) to enhance clarity and visibility. After cold treatment, knockout mutants showed significantly lower survival rates than the wild type, while overexpression lines displayed significantly higher survival (**Figure 5d, e**). To further strengthen the evidence supporting *OsPTR*, we have incorporated additional analyses in line 354-400. In our previous RNA-seq analysis of ten rice accessions, we observed that *OsPTR* expression was significantly higher in accessions with pTE compared to those without pTE (**Figure R1a**). Moreover, *OsPTR* expression was upregulated after 72 hours of cold treatment, suggesting its involvement in the late phase of cold response (**Figure 5b and Supplementary Figure 22**).

Further analysis revealed that pTE is inserted within an intron of *OsPTR* (**Figure 4e**), which may contribute to the production of alternative isoforms and influence overall gene expression^{2,3}. Consistently, Iso-seq data confirmed that the pTE insertion indeed results in transcript isoform differences of *OsPTR* between the two haplotypes (**Figure R1b**). We hope these additional analyses address your concerns and further substantiate the role of *OsPTR* in cold resistance.

Figure 5. The regulation of *OsCACT* and *OsPTR* of cold tolerance in rice. **a, b** Box plot of *OsCACT* (**a**) and *OsPTR* (**b**) expression levels at different cold treatment time points. The different colors represent cold-sensitive (CS) strains, which include six *indica* accessions, and cold-tolerant (CT) varieties, consisting of four *japonica* accessions. Statistical analysis was performed using the Wilcoxon test ($*P < 0.05$, $***P < 0.001$). **c** Box plot of *OsCACT* relative expression levels over time under cold, drought, and salt stress. Statistical analysis was performed using the Student's t-test, with 0h as the reference ($**P < 0.01$, $***P < 0.001$, ns: $P > 0.05$). **d** Representative images of the transgenic lines (*OsCACT* and *OsPTR* knockout mutants and overexpression lines) and their corresponding control of Zhonghua 11 (ZH11) after 5 days of cold treatment followed by 3 days of recovery at the two-leaf and four-leaf stages. Scale bar: 5 cm. **e** Box plot of plant survival rates after 5 days of cold treatment. Statistical analysis was performed using the Student's t-test, with ZH11 as the reference ($*P < 0.05$, $***P < 0.001$). **f** DAB and NBT staining images of leaves from ZH11 and two genes knockout mutants at different cold treatment time points at the two-leaf stage. **g** Line plot of relative electrolyte leakage rates of two genes knockout mutants and ZH11 over the during cold treatment at the two-leaf stage. **h** Box plot of MDA content in ZH11, *OsCACT* knockout mutants, and *OsCACT* overexpression lines after 5 days of cold treatment at the four-leaf stage. Statistical analysis was performed using the Student's t-test, with ZH11 as the reference ($***P < 0.001$, ns: $P > 0.05$). **i** Model diagram of the mechanism by which *OsCACT* enhances cold tolerance in rice. The expression of *OsCACT* in the mitochondrial inner membrane affects the carnitine content in cells, which in turn influences rice cold tolerance through three distinct pathways. Abbreviations: *oscact-ko1*, *OsCACT* knockout mutant-1; *oscact-ko2*, *OsCACT* knockout mutant-2; *OsCACT-OE1*, *OsCACT* overexpression line-1; *OsPTR-OE2*, *OsPTR* overexpression line-2.

OsCACT-OE-2, *OsCACT* overexpression line-2; *osptr-ko*, *OsPTR* knockout mutant; *OsPTR-OE-1*, *OsPTR* overexpression line-1; *OsPTR-OE-2*, *OsPTR* overexpression line-2; DAB, 3, 3'-diaminobenzidine; NBT, nitro blue tetrazolium; MDA, malondialdehyde.

Supplementary Figure 22. Box plot of *OsPTR* relative expression levels over time under cold, drought, and salt stress.

Figure R1. Impact of pTE insertion and transcript variants on *OsPTR* expression.
a, Box plot of *OsPTR* expression level at different cold treatment time points in 10 rice accessions with pTE (+pTE) or without pTE (-pTE). **b**, Iso-seq and RNA-seq reads mapping based on T2T-NIP. The box indicated by the red arrow highlights the difference between the two transcripts.

Figure 4e. ONT reads mapping based on T2T-NIP.

3. As for *OsCACT*, evidence between the gene and cold resistance seems more solid. Yet, the relationship between pTE and *OsCACT* expression is not causally proved, in my opinion.

Response: Thank you for your insightful comment. In our previous RNA-seq analysis of 10 rice accessions, we observed that *OsCACT* expression was significantly lower in accessions with pTE (+pTE) compared to those without pTE (-pTE) before and after cold treatment (**Figure R2a**). Notably, *OsCACT* expression was upregulated after 24 hours of cold treatment, indicating its potential involvement in the early cold stress response.

To explore the mechanism underlying this association, we considered previous studies demonstrating that TE insertions downstream of a gene can influence its transcriptional activity^{4,5}. One plausible explanation is that DNA methylation associated with TE sequences may modulate the expression of nearby genes^{6,7}. To test this hypothesis, we analyzed DNA methylation data at the *OsCACT* locus in *japonica* rice (NIP) which without pTE and *indica* rice (9311) which with pTE. Our results indicate that the pTE insertion in *indica* 9311 is highly methylated (> 80%), whereas no obvious DNA methylation signal (~ 0%) was detected at the *OsCACT* locus in NIP (**Figure R2b**). This differential methylation correlates with the observed downregulation of *OsCACT* expression. Based on these findings, we propose that pTE may modulates *OsCACT* expression through epigenetic modifications.

We have incorporated these findings into the manuscript and revised the corresponding section in the main text (line 354-369) to clarify the relationship between pTE and *OsCACT* expression.

Figure R2. Impact of pTE insertion and DNA methylation on *OsCACT* expression. **a**, Box plot of *OsCACT* expression levels at different cold treatment time points. in accessions with pTE (+pTE) or without pTE (-pTE). **b**, DNA methylation levels before and after cold treatment. The region between the two dashed lines represents the pTE.

4. Overall, authors should be more cautious in their writing and nuance their assertions.

Response: We greatly appreciate the time and effort you have dedicated to reviewing our manuscript and providing detailed and insightful comments. We have carefully revised all statements related to the conclusions as mentioned in the specific comments.

Specific comments:

1) *line 27-29: This is a strong conclusion, not really proved.*

Response: We sincerely thank you for this valuable comment. We have revised the sentence to: “Notably, pTEs exhibited increased H3K27me3 enrichment, suggesting a potential role in epigenetic differentiation under cold stress and transcriptional regulation of the cold response.” We have removed strong terms such as “underscoring” and “critical contribution” and replaced them with more neutral wording to make the conclusion more objective and to avoid overstatement.

2) *l61: any reference for link between TE and SVs?*

Response: Thanks for your helpful suggestion. To address this point, we have added the following relevant references in the revised manuscript⁸⁻¹² in line 63-64.

3) *l 134 ff: are these numbers similar to other reported previously?*

Response: We apologize for the lack of comparison with previous studies. The TE proportions are indeed consistent with previous research^{13,14}, with the TE proportion centered around 51.6%. And we have now added relevant comparisons and citations in the revised manuscript in line 136.

4) *l 211: tone down, 'may suggest' ...*

Response: We appreciate this suggestion to adjust the tone for greater clarity and objectivity. Accordingly, we have revised the sentence in line 218.

Revised sentence:

Moreover, the differences in pTE insertion sites and the conserved patterns between subspecies suggest that pTEs are closely associated with gene expression and species differentiation.

5) *l 218: how were differences in transcription across conditions assessed, is there a statistical test?*

Response: Thank you for pointing out the need for further clarification. We used RNA-seq data for 90 samples, representing 10 materials at three different time points: normal (0h), cold treatment for 24 hours, and cold treatment for 72 hours, with three biological replicates at each time point. After filtering the raw reads, the clean RNA-seq data were aligned to the MH63 reference genome using HISAT2 (v2.2.1). The BAM files and the generated index files were sorted using SAMtools (v1.21) with default parameters. Transcripts were assembled using StringTie (v2.2.0) with default settings. Gene expression was quantified by StringTie, with expression levels reported as fragments per kilobase of transcript per million fragments mapped (FPKM) and transcripts per kilobase million (TPM).

To assess the impact of TE insertions on gene expression, we categorized each gene into two groups based on the presence or absence of TE insertions near the TIP gene, using the TIP map. The gene expression levels were represented by TPM, and the Wilcoxon rank-sum test was used to evaluate whether TE insertions significantly affected gene expression at each time point ($P < 0.05$).

To assess differential gene expression across time points, we used DESeq2 (v1.18.1) and Tetranscripts (v2.2.3) for gene quantification, with the criteria of $|\log_2(\text{Fold Change})| \geq 1$ and an FDR < 0.05 to identify differentially expressed genes (Supplementary Figure 14a). Genes that were differentially expressed at any time point and in any cultivar were classified as cold-responsive genes. Similarly, we used the same method to identify cold-responsive transposable elements (Supplementary Figure 14b). We have added the relevant methods in line 599-609 of the revised manuscript.

Supplementary Figure 14. Evaluation of expression differences across 10 rice accessions under various cold treatment conditions. **a** Differentially expressed genes and TEs identified in each accession under cold treatments (0h vs. 24h and 0h vs. 72h). **b** Expression patterns of all cold-responsive genes and cold-responsive TEs across 10 rice accessions under normal conditions (0h), cold treatment for 24 hours (24h), and cold treatment for 72 hours (72h).

6) *l 224. again, we cannot tell causation from correlation*

Response: We apologize for drawing a conclusion implying causality. In the revised manuscript, we have modified the sentence to: “These findings may suggest that these TE types are likely to play a role in regulating cold tolerance mechanisms in rice.” This

revision clarifies that our conclusion reflects a potential association rather than a definitive causal link.

7) l 231: 'reduced', again, is there a significant test? there are very few samples and many variables, a scenario prone for false positives.

Response: Thank you for raising this important point. We did perform Wilcoxon statistical tests on the expression levels of 11,369 TIP genes as shown in **Figure 2c** and **Supplementary Figure 11**; however, the use of “reduced” was indeed not appropriate. We have revised the sentence to: “We found that pTE insertions within the upstream 2 kb and intronic regions were significantly associated with lower gene expression across all time points, while insertions in exons and downstream 2 kb regions showed no significant association with expression changes.” This revision better reflects the statistical results and provides a more accurate description of the findings.

Figure 2c. The impact of pTE insertions on gene expression levels at different cold treatment time points. pTEs in different rice varieties are categorized into two groups: those with TE (+TE) and those without TE (-TE). “Upstream” indicates that a single pTE is located within 2 kb of the gene and inserted in the upstream region, while “Exon” signifies that a single pTE is also located within 2 kb of the gene but inserted within the exon. Statistical analysis of these data was performed using the Wilcoxon test (***) $P < 0.001$, ns: $P > 0.05$).

Supplementary Figure 11. The impact of pTE insertions on gene expression levels at different cold treatment time points.

8) l 233: *'influence' not proved, -> 'associated with' or similar*

Response: Thanks for your insightful comment. We agree that “influence” may imply a stronger causal relationship. We have revised the sentence to: “In conclusion, pTEs are broadly associated with gene expression in a category- and site-specific manner, with certain TE types and locations exhibiting a more profound impact, especially under stress conditions such as cold treatment.”

9) l 258: *'serve' not proved*

Response: Thank you for pointing this out. We agree that the term “serve” may imply a causal role. To address this concern, we have revised the sentence to: “These results indicate that pTEs may act as potential transcriptional regulatory regions in the rice response to cold stress.” This change softens the interpretation and better reflects the correlative nature of our findings.

10) l 267: *what significance level / criterion was used to determine that a gene is cold responsive?*

Response: Sorry for the lack of clarity in our original description. We calculated the expression levels for each cultivar under normal conditions (0h) and cold stress conditions (24h and 72h). Using the expression levels under normal conditions as a reference, we identified differentially expressed genes (DEGs) at 24h and 72h based on the criteria of $|\log_2(\text{Fold Change})| \geq 1$ and $\text{FDR} < 0.05$. Genes that were differentially expressed at any time point and in any cultivar were classified as cold-responsive genes. Similarly, we used the same method to identify cold-responsive transposable elements (TEs). We have revised this description in the Methods section (line 604-609).

11) l 276: *a co-expression network between TE and coding genes, you mean gene expression and TE genotype?*

Response: Sorry for the lack of clarity in our explanation, which may have caused confusion. The co-expression network refers to the relationship between gene expression and TE expression, not TE genotype. We have made the necessary clarifications in the main text in line 276 and 289-290.

Revised sentence:

Therefore, we analyzed the genome-wide relationship between gene expression and TE expression under cold stress using transcriptome datasets from different cold treatment periods.

To infer the potential biological functions of TEs, we constructed a co-expression network linking TEs with coding genes, based on the correlation between their expression levels.

12) *l 275: GO is really a very confusing world, I suggest to remove but leave that voluntary.*

Response: Thank you for your valuable suggestion. To address this, we have streamlined the relevant content in the main text in line 295-300, removing less relevant details.

Revised sentence:

We then conducted Gene Ontology (GO) enrichment analysis on the genes co-expressed with cis-TE and trans-TE, and found that the functions of these genes are highly associated with the organism's response to abiotic stress. Furthermore, genes co-expressed with both cis-TEs and trans-TEs were significantly related to the non-coding RNA metabolic process (Fig. 3b, Supplementary Fig. 15 and Supplementary Table 8).

13) *l 293-294: how was this done?*

Response: Sorry for the lack of clarity in our previous explanation. The method for identifying transposable elements (TEs) associated with the production of lncRNAs in cis-TEs and trans-TEs is as follows: In the gene and TE co-expression network, TE-gene pairs that show expression correlation are classified based on their location relative to the corresponding genes. TEs located within 100 kb upstream or downstream of a gene are categorized as cis-TEs, while TEs located outside this 100 kb range are classified as trans-TEs. To identify the lncRNA-producing potential of these TEs, transcripts with lengths ≥ 200 nt and open reading frames (ORFs) ≤ 120 amino acids were first screened using ORFfinder. To remove transcripts potentially encoding short peptides, blastx was run with the parameters “-e 1.0e-4 -S 1” against the Swiss-Prot database. Overlaps with rRNA, tRNA, sRNA, and miRNA entries in the Rfam database were also filtered out. Coding potential of the remaining transcripts was evaluated using CPC2¹⁵, PLEK¹⁶, PLncPRO¹⁷ and RNAplnc¹⁸. Only transcripts identified as non-coding by at least two of these tools were classified as lncRNAs. Finally, to identify TE-derived lncRNAs, the transcripts classified as lncRNAs were overlapped with MH63 TE annotations to produce a set of TE-derived lncRNAs. Overall, we identified 337 TEs associated with production of lncRNA in cis-TEs and 10,755 in trans-TEs. We have provided a detailed description of this method in the Methods section (line 639-648).

14) *l 305 how were these pTE chosen?*

Response: Thank you for pointing out this issue. Upon review, we found that “pTE” was indeed a typo and should have been “TE”. We apologize for this mistake, and the correction has been made in the manuscript in line 305. The TE chosen method was as follows: First, we constructed a gene-TE co-expression network using four software tools: WGCNA, Pearson correlation coefficient (PCC), CORNET, and TEffectR. To ensure the robustness of the results, co-expression relationships were retained only if detected by at least two software tools. We then selected 1,529 TEs co-expressed with 138 cold-responsive rice genes. These steps are described in detail in the Methods section (line 649-678).

15) l 306. *why lncRNA?*

Response: Thank you for raising this issue. Previous studies¹⁹ have found that expressed TEs influence the host gene regulatory network primarily through the production of regulatory ncRNAs and the co-option of TE-derived coding sequences as new transcriptional effector proteins. Based on these findings, we focused on analyzing the potential role of TE-derived lncRNAs in rice cold response. Apologies for not providing a clearer connection with the previous sections. We have made revisions in line 300-304 of the manuscript to improve the logical flow.

Revised sentence:

Consistent with previous studies suggesting that TEs influence host gene networks through long non-coding RNAs (lncRNAs) or TE-derived transcriptional regulators, we identified 337 TEs associated with production of lncRNA in cis-TEs and 10,755 in trans-TEs, which may have an impact on rice cold tolerance.

16) l 319: *why different genes here and in previous list (l 295)*

Response: Thank you for pointing out this inconsistency. We apologize for the oversight regarding the gene lists. In the revised manuscript, we have corrected this section and revised **Supplementary Table 11** to ensure consistency.

Revised sentence:

To investigate the impact of pTEs on cold tolerance in rice, we analyzed 138 previously studied cold-responsive genes (**Supplementary Table 9**) and identified 54 genes containing pTEs (**Supplementary Table 11**).

17) l 350: *survival rate of both genotypes by stage*

Response: Thank you for pointing out this issue. We have now included the relevant data and revised the manuscript in line 386-389.

Revised sentence:

After 5 days of cold treatment at 4°C, the *osptr* showed less leaf curling than ZH11, with an average survival rate of 15%, significantly lower than the 91% survival rate recorded in ZH11 at the two-leaf stage.

18) l 587: are the 90 samples the same set as mentioned that correspond to 10 different genotypes only? Not clear how TE expression is measured

Response: We apologize for the lack of clarity in our previous description. The 90 RNA-seq samples are consistent with the assembled 10 rice genomes, representing 10 cultivars subjected to 6 to 8°C (dark/light) treatment at 0h (normal conditions), 24h, and 72h, with three biological replicates at each time point. We have provided a description of the 90 samples in the Methods section (line 584-586). Regarding the TE expression calculation method: we used TEtranscripts (v2.2.3) to compute the expression matrix for TEs. Differential expression analysis of TEs was performed on the raw count matrix using DESeq2 (v1.18.1). A detailed description of this process is provided in line 599-609 of the Methods section.

19) l 603 : why that value (0.55)

Response: Thank you for your question. It is well established that a correlation coefficient between 0.4 and 0.69 is considered indicative of a moderate correlation, a widely accepted criterion²⁰. Moreover, Aoki et al.²¹ demonstrated that a PCC threshold of 0.55 effectively identifies biologically meaningful modules. Consequently, we adopted this threshold for our analysis and have cited the reference in the revised manuscript (line 667) to offer further context.

20) l 618: what is ndG?

Response: Thank you for raising this question. We have clarified the full term in line 682 of the main text. ndG (normalized deltaG) is a novel parameter introduced in the LncTar tool²² as a scoring metric for predicting LncRNA-mRNA interactions. It reflects the binding affinity between an LncRNA and its target mRNA. Given that longer RNA molecules generally exhibit lower dG values, using raw dG values as a criterion for RNA-RNA interaction may not be suitable. By normalizing dG to the lengths of the RNA molecules, ndG provides a more rational assessment. The calculation formula for ndG is as follows:

$$ndG = dG / \min(\text{length}_{\text{lncRNA}}, \text{length}_{\text{mRNA}})$$

where $\text{length}_{\text{lncRNA}}$ and $\text{length}_{\text{mRNA}}$ represent the lengths of the candidate lncRNA and the other RNA sequence, respectively. ndG is a floating-point value that can serve as a threshold to determine whether two RNA molecules interact. LncTar considers two RNA molecules to interact if the computed ndG value is less than or equal to the specified ndG threshold (e.g., -0.05)²³. Lower ndG values indicate stronger binding affinities, making the corresponding mRNA potential targets of the lncRNA.

21) l 621ff: what GWAS is done? what is the phenotype?

Response: Thank you for pointing out the lack of clarity in our description. In this study, we conducted genome-wide association study (GWAS) using the mixed linear model (MLM) approach implemented in EMMAX²⁴. The phenotype under investigation was the survival rate of 165 rice seedlings subjected to 5 days of cold treatment followed by 3 days of recovery. These 165 rice cultivars represent the major

rice subpopulations globally, encompassing 14 *aus*, 23 *indica* I, 9 *indica* II, 27 *indica* III, 21 *indica* intermediate, 3 intermediate, 12 *japonica* intermediate, 27 temperate *japonica*, 28 tropical *japonica*, and 1 VI/aromatic. To evaluate the cold tolerance of these selected rice cultivars, 30-day-old seedlings were placed in a growth chamber maintained at 6 to 8 °C (dark/light) with 12,000 Lux of artificial light (GC400/230NG, Gavita, China) and 70% relative humidity for 5 days. Following this, the plants were transferred to room temperature for 3 days, and their phenotypic responses were documented, as detailed in **Supplementary Table 6**. We have revised the manuscript to address this issue and ensure clarity in line 691-700.

22) l 672: *what is the metabolomic analysis done for?*

Response: Thank you for raising this question. As stated in the main text (line 409), the metabolomic analysis was performed to examine changes in carnitine content (**Supplementary Figure 25**), in order to explore the potential role of *OsCACT* in the cold stress response. Compared to the wild-type ZH11, the mutant lines exhibited a significant decrease in the abundance of Levocarnitine and O-Acetylcarnitine. Moreover, under cold stress conditions, both the mutant lines and ZH11 exhibited markedly low levels of these metabolites. These results suggest that *OsCACT* may enhance cold stress tolerance by modulating fatty acid metabolism.

Supplementary Figure 25. The content of Levocarnitine and O-Acetylcarnitine in leaves of ZH11 and *OsCACT* knockout mutants. Statistical analysis was performed using the Wilcoxon test (***) $P < 0.001$.

Minor comments

23) *k* symbol for thousand is lower case, they also use *K* for number of clusters in Admixture.

Response: Thank you for pointing this out. We appreciate the clarification regarding the use of the lowercase “k” for thousand and the uppercase “K” for the number of

clusters in Admixture. In the revised manuscript, we have ensured consistency and adhered to this convention to avoid confusion.

Reviewer #2 (Remarks to the Author):

Transposable elements (TEs) are ubiquitous genomic components and play important roles in some traits variations by affecting the expression of their nearby genes in plants. Although an increasing number of rice studies focused on identifying TEs in rice populations and constructing pan-TE maps, which have revealed numerous TE characteristics related to rice domestication and agronomic traits, the influence of TEs on cold tolerance in rice remains largely unexplored. In this study, the authors constructed a pangenome graph and identified 30,316 TIP sites, highlighting significant diversity among pTEs. Elaborated on the contribution of pTEs differential gene expression prior to and following cold stress, and identified numerous cold-responsive genes and pTEs. TIP-GWAS on cold tolerance phenotypes in 165 rice accessions led to the discovery of novel cold-tolerance genes, such as OsCACT and OsPTR, and overexpression experiments and metabolomic data was validated. Overall, the logic is good, the analyses are clear and data are reliable. Here are some comments:

Response: We thank the reviewer for carefully reading our manuscript, and giving us the valuable suggestions and comments.

1. *Many errors/typos in the manuscript, either from the text of or from the figures, need to carefully checked and revised. Here are only several examples.*

(1) *Some specialized nouns should be written properly. “long interspersed nuclear element (LINE)” in Line 72 should be “long interspersed nuclear elements (LINEs)”, Stow should be Stowaway in Line 339 and Figure 1d. The authors could follow Wicker et al, 2007 to correct the naming system of transposable elements in the manuscript. Authors are kindly requested to carefully review the entire text.*

Response: Thank you for your thorough review and for pointing out these errors. We have revised “LINE” to “LINES” in line 75 and carefully reviewed the entire manuscript. Following the classification system proposed by Wicker et al.²⁵, we have standardized the TE classifications across all main and supplementary figures. Specifically, we corrected “Stow” to “Stowaway”, “MLE” to “Tc1-Mariner”, “MULE” to “Mutator”, and “PILE” to “PIF-Harbinger”. These updates have been implemented to ensure consistency and accuracy in the terminology used throughout the manuscript.

(2) *Some specialized nouns should be italicized, such as *Oryza sativa* in Line 433 and *P* value in Figure 2 and in Figure 5, etc. Authors are kindly requested to carefully review the entire text.*

Response: Thank you for highlighting this issue. We appreciate your careful review and have gone through the entire manuscript to ensure proper formatting of specialized nouns, such as italicizing *Oryza sativa* and correcting the format of *P*-value in Figure 2, Figure 5, and elsewhere. All necessary adjustments have been made in the revised manuscript.

2. Why wasn't the MITEs family included in Fig1b? It would be better to add MITEs in Figure 1b because the MITEs is also a major part of TEs.

Response: Thank you for pointing out this issue. We agree that MITEs are indeed an important component of TEs. To address this, we have updated the classification method to include the MITE family while ensuring the overall proportion of TEs remains consistent. This update is reflected in the revised **Figure 1b**.

Figure 1b. TE content across the 10 rice assemblies. The phylogenetic tree was constructed based on single-copy genes from the assemblies.

3. Line 157. The obtained TIPs, particularly the heterozygous TIPs, should be validated with more evidences, e.g., the PCR experiments or the IGV browser. And their accuracy rate could be estimated.

Response: We sincerely appreciate this valuable comment. First, we randomly selected several TIPs and examined the read breakpoints at specific loci using IGV to validate their accuracy. Additionally, we employed SV calling methods based on long reads to assess the overall accuracy of the 30,316 TIPs. The detailed methodology is as follows:

To validate the accuracy of the TIPs, we randomly selected 8 TIPs and performed detailed checks using the IGV browser²⁶. During the validation process, for SVs with deletions in *indica* rice, we used *japonica* rice T2T-NIP²⁷ as the reference genome, and for deletions in *japonica* rice, we used *indica* rice MH63²⁸ as the reference genome. The length, position, and genotyping of these TIPs were validated using reads (**Supplementary Figure 6**), further supporting the accuracy of our TIPs. We have revised the manuscript in line 174-176 and added a new **Supplementary Figure 6** to reflect these updates.

Since TIPs are a subset of SVs, and our SVs were identified using minigraph, we employed another SV calling method based on long reads to obtain an independent SV set (**Figure R3**). This allowed us to validate the consistency between the two SV identification methods and further confirm the accuracy of TIPs. We used the PanPop pipeline²⁹, which integrates four SV-calling tools: Sniffles³⁰, cuteSV³¹, svim³², and pbsv (<https://github.com/PacificBiosciences/pbsv>), and thus provides more reliable

results. We identified 50,875 SVs using minigraph, among which 47,592 (93.5%) were consistent with the results from PanPop. Of these SVs, 30,316 were identified as TIPs, with 28,590 (94.3%) showing consistency with the PanPop results (**Figure R3**), indicating the high accuracy of the obtained TIPs.

Supplementary Figure 6. Validation of TIP genotyping based on read mapping.

Figure R3. Validation of SVs and TIPs by two pipelines.

4. Line 174-175. There is no figure or table that can support “pTE insertion rates were significantly higher in 4 assemblies, with the peak primarily contributing to TEs shared by the 4 japonica assemblies”. In general, when we say “significant”, it need to be based on statistical indicators, such as a p-value less than 0.05 or 0.01.

Response: We apologize for not providing a reference to the supplementary figure in the original manuscript. To address this, we have added a citation to **Supplementary Figure 7** in line 178, which illustrates that pTE insertion rates are visibly higher in the four *japonica* assemblies, as indicated by the blue bars in the plot. However, since each assembly corresponds to a single data point, statistical analysis could not be performed. Therefore, to avoid overstatement, we have removed the term “significantly” from the revised manuscript.

Supplementary Figure 7. The distribution of gene-flanking pTEs, genome-wide pTEs, and genome-wide SVs.

5. Line 186-191. The authors declared that “the distribution suggested a preference for pTE insertion in the upstream 2 kb 191 region” because 9,131 events were identified within the 2kb region and only 7,559 occurred within gene bodies. Typically, when

comparing the frequency of TEs, it is necessary to use their density rather than the total number to avoid an incorrect comparison due to different sequence lengths.

Response: We are grateful for your valuable feedback on this issue. You are absolutely right that using insertion density rather than total counts is a more appropriate and accurate approach, as it accounts for differences in region lengths and avoids potential bias. In response, we recalculated the pTE insertion densities across different sequence regions and updated **Supplementary Figure 8** accordingly. Our revised analysis, based on insertion density per Mb, shows that the frequency of pTE insertions is highest in the upstream 2 kb region, nearly double that of the gene body, supporting our previous conclusion. We have revised the manuscript in line 195-198 to: “In general, the pTE insertion densities indicated a preference for insertion in the upstream 2 kb region, potentially influencing gene expression through the regulation of promoter activity.” In addition, the density analysis for MITE/Tourist-type pTEs further supports the conclusion that: “MITE/*Tourist* pTEs exhibited a strong preference for insertion in the 2 kb regions flanking genes rather than within gene bodies.” We have made the necessary revisions to the manuscript to reflect these changes.

Supplementary Figure 8. Bar plot showing the pTE insertion densities across different sequence regions.

6. For TIP-eQTL and TIP-GWAS, whether the TIPs absent in the reference genome are used or not? If not, would this introduce a bias towards the reference genome?

Response: Thank you for raising this important question. In our analysis, we utilized SV calling by minigraph³³, which enabled the detection of SVs that are absent in the reference genome. These SVs were then annotated with TE information to construct the TIP map. For instance, although the TIP identified within *OsPTR* is not present in the

reference genome (**Figure 4e**), it was included in downstream analyses. Consequently, our TIP dataset includes TIPs absent in the reference genome, ensuring a comprehensive analysis and minimizing potential bias towards the reference genome.

Figure 4e. ONT reads mapping based on T2T-NIP. The top plots display the gene structure diagrams. The four tracks labeled K, R, TB, and L represent haplotype A (HapA) for *japonica*, while the six tracks labeled KO, M, N, NJ, T, and Y represent haplotype B (HapB) for *indica*. The numbers at the right end of each track indicate the reads coverage, and the area between the two dashed lines represents the deletion.

7. Line 456-457, “these contigs were aligned to a reference genome”. The reference genome should be clearly indicated. Why was it chosen as the reference genome? This is important because different reference genomes can result in variations in genome size and sequence using RagTag.

Response: We apologize for the lack of clarity in our description. As rice is divided into two subspecies, *indica* and *japonica*, we chose the reference genomes according to the subspecies classification of our materials. For the *indica* materials, we used the MH63 gap-free genome²⁸, and for the *japonica* materials, we used the telomere-to-telomere genome NIP²⁷ for alignment. This choice of reference genomes was made to ensure the accuracy of the alignment and to better reflect the genetic diversity within the rice subspecies. We have updated the manuscript to clearly indicate these reference genomes in line 507-508.

8. Fig4.f-g. Since *japonica* is inherently more cold-tolerant than *indica*, it would be better to provide both the phenotype and the gene expression data to support the functional differences among different haplotypes of the two genes.

Response: We appreciate your insightful suggestion. In response, we have included both the phenotype and gene expression data in **Supplementary Figure 19** and have revised the corresponding statement in line 354-356 of the manuscript to better reflect significant differences in survival rates among the different haplotypes after cold stress.

Supplementary Figure 19. Phenotypic and transcriptional differences of *OsCACT* and *OsPTR* between *japonica* and *indica* haplotypes under cold stress.

a, Comparison of survival rates for different haplotypes of *OsCACT* and *OsPTR* under cold stress. **b**, Comparison of gene expression for different haplotypes of *OsCACT* and *OsPTR* under cold stress. Statistical analysis was performed using the Wilcoxon test (** $P < 0.01$, *** $P < 0.001$).

9. Line 1187. Please add the accession numbers of the 10 rice assemblies and their annotation (including TE annotation and gene annotation) in NCBI or other public databases in the Data availability section.

Response: Thank you for pointing this out. We apologize for the omission. We have submitted the 10 rice assemblies along with their gene annotations in National Genomics Data Center (NGDC; <https://ngdc.cncb.ac.cn/>) under accession number PRJCA032145. Their TE annotations are available at Rice pTE Database (<https://cbi.gxu.edu.cn/RICEPTEDB/>). We have updated the Data Availability section accordingly in line 1324.

Reviewer #3 (Remarks to the Author):

Here, the authors assembled 10 high-quality rice genomes by various sequencing methods and constructing a pangenome graph; Using the pangenome data and resequencing 165 rice accessions, the authors identified 30,316 transposable element insertion polymorphism (TIP) sites; at the same time, they identified TEs responsive to cold stress from transcriptome data; at last using the data TIP-GWAS revealed novel cold-tolerant genes, *OsCACT* and *OsPTR*. The manuscript provided the valuable pTEs data and valuable targets for breeding cold-tolerant rice varieties. However, some other rice pan-genome studies had many *de novo* assembled genomes include many varieties of the selected 10 rice accessions, such as Koshihikari, Lemont, Nipponbare, 9311, etc. In addition, why did transposable element insertion or deletion of the two novel cold-tolerant gene change their expression?

Response: We thank the reviewer for carefully reading our manuscript, and giving us the valuable suggestions and comments.

1. Since the other rice pan-genome studies had many *de novo* assembled genomes, the authors should construct the rice pangenome graph by integrating these genomes, and

construct a bigger pan-TEs.

Response: Thank you for your insightful comment. We selected the 10 genomes because they represent materials with well-characterized levels of cold tolerance and exhibit significant differences in their cold tolerance phenotypes. Additionally, previous studies have shown that when comparing a sample to a pangenome, variants present in the pangenome but absent from the sample can lead to misleading results and ambiguity of the reference genome, such as false read mappings^{34,35}. Therefore, our study primarily focuses on these 10 genomes.

Nonetheless, to enhance the utility and applicability of the pangenome, we incorporated published rice genomes and constructed an expanded pan-TE dataset following the workflow outlined in **Figure 2a**. It is worth mentioning that Shang et al. published 202 rice genomes¹⁴. Although these genomes represent extensive diversity, they lack chromosome-level assemblies and were therefore excluded from our analysis. This exclusion is due to the fact that minigraph, the tool we used to construct the pangenome graph, is optimized to work better with chromosome-level assemblies.

Specifically, we collected a total of 131 published high-quality rice genomes (**Table R1**). Combined with our assembled 10 genomes, we constructed a pangenome graph using MH63 as the backbone with minigraph. The total size of the graph is 986.8 Mb, containing 142,666 SVs. Based on the workflow in **Figure 2a**, these SVs were annotated, resulting in a total of 35,676 TIPs, with 103.4 Mb annotated as TEs. We have updated the relevant description in the main text (line 223-226) and made the data publicly available through the Rice pTE Database (<https://cbi.gxu.edu.cn/RICEPTEDB/>).

Table R1. Summary of 131 rice sample information.

Sample ID	Variety name	Reference
TG1	AC 13 (T 141)::IRGC 5456-1	Zhang et al., 2022 ³⁶
TG2	91-382::IRGC 63464-1	Zhang et al., 2022 ³⁶
TG3	ARC 11768::IRGC 21630-1	Zhang et al., 2022 ³⁶
TG4	MAK BOUAP::IRGC 30106-3	Zhang et al., 2022 ³⁶
TG5	BUCAYAB::IRGC 44357-1	Zhang et al., 2022 ³⁶
TG6	ARC 11777::IRGC 21639-1	Zhang et al., 2022 ³⁶
TG7	BANDI::IRGC 17214-1	Zhang et al., 2022 ³⁶
TG8	DERAWA::IRGC 64106-1	Zhang et al., 2022 ³⁶
TG9	HERATH BANDA::IRGC 67630-1	Zhang et al., 2022 ³⁶
TG10	IH PEN SHIM MING::IRGC 26067-1	Zhang et al., 2022 ³⁶
TG11	JHONA 101::IRGC 27976-1	Zhang et al., 2022 ³⁶
TG12	KASHA::IRGC 83865-1	Zhang et al., 2022 ³⁶
TG13	LUO SI ZHAN::IRGC 67211-1	Zhang et al., 2022 ³⁶
TG14	MALAGKIT (PINELIPE)::IRGC 67444-1	Zhang et al., 2022 ³⁶
TG15	BUKU::IRGC 28683-2	Zhang et al., 2022 ³⁶
TG16	PHAN PHAE::IRGC 29652-2	Zhang et al., 2022 ³⁶
TG17	PLE LIA::IRGC 73562-1	Zhang et al., 2022 ³⁶

TG18	SHANGYIPA::IRGC 64928-1	Zhang et al., 2022 ³⁶
TG19	Qingjinzaosheng	Zhang et al., 2022 ³⁶
TG21	Kahamu	Zhang et al., 2022 ³⁶
TG22	Qiutianxiaoting	Zhang et al., 2022 ³⁶
TG24	Lucaihao	Zhang et al., 2022 ³⁶
TG27	HODARAWALA::IRGC 67631-1	Zhang et al., 2022 ³⁶
TG28	LALKA (LAL DHAN)::IRGC 64946-1	Zhang et al., 2022 ³⁶
TG29	MAEKJO::IRGC 77666-1	Zhang et al., 2022 ³⁶
TG30	MUKKALA BAZAL::IRGC 77279-1	Zhang et al., 2022 ³⁶
TG31	SAHULO FACHE SOYO::IRGC 66630-1	Zhang et al., 2022 ³⁶
TG32	SAL BUI BAO	Zhang et al., 2022 ³⁶
TG33	Tun Sart	Zhang et al., 2022 ³⁶
TG34	UP 15	Zhang et al., 2022 ³⁶
TG42	Cisadane	Zhang et al., 2022 ³⁶
TG43	Khao Dawk Mali 105	Zhang et al., 2022 ³⁶
TG45	TAICHUNGNATIVE1	Zhang et al., 2022 ³⁶
TG46	MR 19	Zhang et al., 2022 ³⁶
TG49	DV 86::IRGC 8840-1	Zhang et al., 2022 ³⁶
TG50	ITA 117::IRGC 75235-1	Zhang et al., 2022 ³⁶
TG51	MBEIMBEIHUN::IRGC 69730-1	Zhang et al., 2022 ³⁶
TG52	VARIRANGAHY::IRGC 69897-1	Zhang et al., 2022 ³⁶
TG53	ZINYA KOLAMBA::IRGC 52402-1	Zhang et al., 2022 ³⁶
TG54	ARC 18578::IRGC 42459-2	Zhang et al., 2022 ³⁶
TG55	BR 116-3B-53::IRGC 39559-2	Zhang et al., 2022 ³⁶
TG56	E 21::IRGC 33929-2	Zhang et al., 2022 ³⁶
TG58	LUDI GOCHYA::IRGC 26504-2	Zhang et al., 2022 ³⁶
TG59	OR 117-8::IRGC 39680-2	Zhang et al., 2022 ³⁶
TG60	AMAKOYALI::IRGC 60878-1	Zhang et al., 2022 ³⁶
TG61	GAM PERNG::IRGC 62135-1	Zhang et al., 2022 ³⁶
TG62	HIJOL DIGA::IRGC 31655-1	Zhang et al., 2022 ³⁶
TG63	SOM::IRGC 92221-1	Zhang et al., 2022 ³⁶
TG64	KAUK PAHLING::IRGC 95823-2	Zhang et al., 2022 ³⁶
TG65	American Huangkedao	Zhang et al., 2022 ³⁶
TG68	Guangluai 4	Zhang et al., 2022 ³⁶
TG70	Minbeiwaxian	Zhang et al., 2022 ³⁶
TG75	Annongwangeng B	Zhang et al., 2022 ³⁶
TG76	IRGA 318-11-6-9-2B::IRGC 117339-1	Zhang et al., 2022 ³⁶
TG77	WAS 173-B-B-6-2-2::C1	Zhang et al., 2022 ³⁶
TG78	KHAO GRADOOK CHAHNG::IRGC 17111-1	Zhang et al., 2022 ³⁶
TG80	Gang 46B	Zhang et al., 2022 ³⁶
TG81	Binam	Zhang et al., 2022 ³⁶
TG82	X 21	Zhang et al., 2022 ³⁶
TG83	Budda	Zhang et al., 2022 ³⁶
TG84	Tek Si Chut	Zhang et al., 2022 ³⁶

TG85	Karnal Local	Zhang et al., 2022 ³⁶
TG86	Chorofa	Zhang et al., 2022 ³⁶
TG87	KASALATH	Zhang et al., 2022 ³⁶
TG88	PSBRC 80	Zhang et al., 2022 ³⁶
TG90	Jinyuan 85	Zhang et al., 2022 ³⁶
WW8	JADO::IRGC 61966-1	Zhang et al., 2022 ³⁶
WSSM	Wushansimiao	Zhang et al., 2022 ³⁶
QUAN	Quan9311	Zhang et al., 2022 ³⁶
H7L1	Huanghuazhan	Zhang et al., 2022 ³⁶
H7L26	CDR22	Zhang et al., 2022 ³⁶
H7L27	PsBRC28	Zhang et al., 2022 ³⁶
H7L28	PsBRC66	Zhang et al., 2022 ³⁶
H7L29	IR64	Zhang et al., 2022 ³⁶
H7L30	Teqing	Zhang et al., 2022 ³⁶
H7L31	IR50	Zhang et al., 2022 ³⁶
H7L32	OM1723	Zhang et al., 2022 ³⁶
H7L33	Phalguna	Zhang et al., 2022 ³⁶
SE-3	BR 24	Zhang et al., 2022 ³⁶
SE-19	Zhong 413	Zhang et al., 2022 ³⁶
SE-33	BG 300	Zhang et al., 2022 ³⁶
SE-134	Haonnong	Zhang et al., 2022 ³⁶
DHX2	Daohuaxiang2hao	Qin et al., 2021 ¹³
02428	02428	Qin et al., 2021 ¹³
Kosh	Koshihikari	Qin et al., 2021 ¹³
ZH11	Zhonghua11	Qin et al., 2021 ¹³
KY131	Kongyu131	Qin et al., 2021 ¹³
Lemont	Lemont	Qin et al., 2021 ¹³
NamRoo	Nam Roo	Qin et al., 2021 ¹³
LJ	Lijiang Xintuan Hei Gu	Qin et al., 2021 ¹³
G46	Gang46	Qin et al., 2021 ¹³
CN1	Chuannong1	Qin et al., 2021 ¹³
FS32	FS32	Qin et al., 2021 ¹³
DG	Digu	Qin et al., 2021 ¹³
D62	D62	Qin et al., 2021 ¹³
II32	II32	Qin et al., 2021 ¹³
R527	R527	Qin et al., 2021 ¹³
S548	Shuhui548	Qin et al., 2021 ¹³
9311	9311	Qin et al., 2021 ¹³
Y58S	Y58S	Qin et al., 2021 ¹³
J4155	J4155	Qin et al., 2021 ¹³
G8	Guang8	Qin et al., 2021 ¹³
Y3551	Yihui3551	Qin et al., 2021 ¹³
IR64	IR64	Qin et al., 2021 ¹³
TM	Tsipala Menahar	Qin et al., 2021 ¹³

TUMBA	Tumba	Qin et al., 2021 ¹³
G630	Gui630	Qin et al., 2021 ¹³
YX1	Yixiang1	Qin et al., 2021 ¹³
WSSM	Wushansimiao	Qin et al., 2021 ¹³
FH838	Fuhui838	Qin et al., 2021 ¹³
N22	Nagina22	Qin et al., 2021 ¹³
Basmati1	Basmati1	Qin et al., 2021 ¹³
XL628S	Xiangling628S	Zhang et al., 2022 ³⁷
LK638S	Longke638S	Zhang et al., 2022 ³⁷
J4155S	Jing4155S	Zhang et al., 2022 ³⁷
HuaZhan	Huazhan	Zhang et al., 2022 ³⁷
NIP-T2T	Nipponbare	Shang et al., 2023 ²⁷
MH63RS3	Minghui63	Song et al., 2022 ²⁸
ZS97RS3	Zhenshan97	Song et al., 2022 ²⁸
OsCMeo	CHAO MEO::IRGC 80273-1	Zhou et al., 2020 ³⁸
OsAzu	Azucena	Zhou et al., 2020 ³⁸
OsKeNa	KETAN NANGKA::IRGC 19961-2	Zhou et al., 2020 ³⁸
OsARC	ARC 10497::IRGC 12485-1	Zhou et al., 2020 ³⁸
OsPr106	PR 106::IRGC 53418-1	Zhou et al., 2020 ³⁸
OsIR64	IR 64	Zhou et al., 2020 ³⁸
OsLima	LIMA::IRGC 81487-1	Zhou et al., 2020 ³⁸
OsKYG	KHAO YAI GUANG::IRGC 65972-1	Zhou et al., 2020 ³⁸
OsGoSa	GOBOL SAIL (BALAM)::IRGC 26624-2	Zhou et al., 2020 ³⁸
OsLixu	LIU XU::IRGC 109232-1	Zhou et al., 2020 ³⁸
OsLaMu	LARHA MUGAD::IRGC 52339-1	Zhou et al., 2020 ³⁸
OsNaBo	NATEL BORO::IRGC 34749-1	Zhou et al., 2020 ³⁸

2. *Supplementary Table 1. The cold-tolerance level of the 10 rice accessions is not even, it need add some accessions on cold treatment for RNA-seq analysis.*

Response: Thank you for your valuable suggestion. Cold-tolerance levels were determined based on survival rates. Following cold treatment, the survival rates of the 165 accessions were predominantly distributed within the ranges of 0-10% and 90-100% (**Figure R4**), consistent with previous study³⁹. Due to the limited number of accessions exhibiting intermediate tolerance, selecting even cold-tolerance level accessions was challenging. To avoid potential ambiguity, the cold-tolerance classification has been removed from **Supplementary Table 1** and replaced with the corresponding survival rates after cold treatment.

Regarding the suggestion to include additional accessions for RNA-Seq analysis, we fully acknowledge its significance and agree with this recommendation. Accordingly, 6 representative accessions were selected from the original set of 165 for RNA-seq analysis (**Table R2**). The results consistently confirmed our initial findings of the 10 rice accessions, showing the same expression pattern before and after 24-hour cold treatment (**Figure R5**). These findings further support the association between pTE presence and the regulation of *OsPTR* and *OsCACT* expression. We have updated

the manuscript and supplementary information accordingly with all relevant data in line 354-377.

Figure R4. Density plot of survival rates of 165 rice accessions after cold treatment.

Supplementary Table 1. Summary of 10 rice accessions.

Name	Accession	Subpopulation	Country of origin	Survival rate (%)
KOGONI 91-1::C1	KO	indica intermediate	Mali	0
Koshihikari	K	temperate japonica	Japan	100
Lemont	L	tropical japonica	USA	52
MADINIKA 1329::GERVEX 8366-C1	M	indica III	Madagascar	0
Nanjing11	NJ	indica I	China	0
NONA_BOKRA	N	indica intermediate	India	0
Nipponbare	R	temperate japonica	Japan	100
Fujisaka5	TB	temperate japonica	Japan	100
TEQING	T	indica intermediate	China	0
9311	Y	indica II	China	0

Table R2. Summary of 6 rice accessions.

Sample_name	OsCACT haplotype	OsPTR haplotype	Survival rate (%)	Cultivar
L_0261	+pTE	-pTE	0±0	KITRANA 1007
L_0303	-pTE	+pTE	100±0	IR 73688-57-2
L_0399	+pTE	-pTE	0±0	R 582
L_0408	+pTE	-pTE	0±0	XIANG CHANG ZAO
L_0578	-pTE	+pTE	100±0	PLOVDIV_24
L_0589	-pTE	+pTE	100±0	DANGO MOCHI

Figure R5. Gene expression levels of *OsPTR* and *OsCACT*. **a**, Box plot of *OsPTR* expression levels at different cold treatment time points in 10 rice accessions. **b**, Box plot of *OsPTR* expression levels at different cold treatment time points in 6 additional rice accessions. **c**, Box plot of *OsCACT* expression levels at different cold treatment time points in 10 rice accessions. **d**, Box plot of *OsCACT* expression levels at different cold treatment time points in 6 additional rice accessions. Statistical analysis was performed using the Wilcoxon test (* $P < 0.05$, ** $P < 0.01$, *** $P < 0.001$).

3. *Fig4*, TIP-GWAS revealed novel cold-tolerant genes, *OsCACT* and *OsPTR*, It is not clear that the transposable element insertion or deletion induce or reduce the expression of *OsCACT* and *OsPTR*; it need some data to support the results.

Response: Thank you for the valuable comments. In our previous RNA-seq analysis of 10 rice accessions, we observed that in samples with a pTE insertion downstream of *OsCACT*, the gene expression is significantly downregulated (**Figure R2**). In light of previous studies suggesting that TE insertions near genes can cause downregulation^{4,5}, possibly due to DNA methylation^{6,7}, we compared DNA methylation levels at the *OsCACT* locus between *japonica* rice (NIP, without pTE) and *indica* rice (9311, with pTE). The data reveal that the pTE insertion in *indica* 9311 is highly methylated, correlating with the observed reduction in *OsCACT* expression, thereby supporting the

hypothesis that pTE may modulates *OsCACT* expression through epigenetic mechanisms.

For *OsPTR*, our RNA-seq data showed that a significant expression difference between the two haplotypes (with and without pTE) is evident after cold treatment (**Figure R1a**). Further analysis indicated that the pTE is inserted within an intron of *OsPTR* (**Figure 4e**), which may lead to the production of different isoforms and thereby alter overall gene expression^{2,3}. Consistently, Iso-seq data confirmed that the pTE insertion indeed results in transcript isoform differences of *OsPTR* between the two haplotypes (**Figure R1b**). These data indicate that the pTE insertion within *OsPTR* may alter the gene structure, potentially contributing to the observed difference in gene expression.

In summary, the integrated evidence from RNA-seq, DNA methylation, and Iso-seq analyses indicates that pTE insertions influence the expression of both *OsCACT* and *OsPTR*, likely through epigenetic regulation and alternative isoform generation, thereby contributing to enhanced cold tolerance in rice. We have updated the manuscript accordingly with all relevant data in line 354-377.

Figure R2. Impact of pTE insertion and DNA methylation on *OsCACT* expression. **a**, Box plot of *OsCACT* expression levels at different cold treatment time points. in accessions with pTE (+pTE) or without pTE (-pTE). **b**, DNA methylation levels before and after cold treatment. The region between the two dashed lines represents the pTE.

Figure R1. Impact of pTE insertion and transcript variants on *OsPTR* expression.

a, Box plot of *OsPTR* expression level at different cold treatment time points in 10 rice accessions with pTE (+pTE) or without pTE (-pTE). **b**, Iso-seq and RNA-seq reads mapping based on T2T-NIP. The box indicated by the red arrow highlights the difference between the two transcripts.

Figure 4e. ONT reads mapping based on T2T-NIP.

4. Because *ZH11* (temperate japonica) is cold-tolerant rice variety. Except checking function of *OsCACT* and *OsPTR* in *ZH11*, adding a cold-intolerant variety is better.

Response: Thank you for your valuable and insightful comments. In response to your suggestion, we selected the cold-sensitive *indica* variety Huanghuazhan (HHZ) as the genetic background. We amplified the full-length coding sequence (CDS, excluding the stop codon) of *OsCACT* using cDNA from the *japonica* variety *ZH11* as the template, and cloned the PCR product into the pRHVNGFP vector⁴⁰. The resulting construct was introduced into HHZ via *Agrobacterium tumefaciens*-mediated transformation. qRT-PCR analysis confirmed the overexpression of *OsCACT* in the transgenic plants (Supplementary Figure 27a). We then grew both wild-type (WT) and T0 generation overexpression lines (*OsCACT*-OE in HHZ background) to the four-leaf stage, subjected them to cold treatment at 8–10°C for 5 days, followed by a 3-day recovery period. *OsCACT*-OE (HHZ) lines exhibited enhanced cold tolerance compared to WT plants (Supplementary Figure 27b, c). These results indicate that overexpression of *OsCACT* improves cold tolerance in rice.

Unfortunately, the transformation of *OsPTR* into HHZ was not successful despite multiple attempts. This may be attributed to the generally low transformation efficiency in *indica* rice varieties, as well as the presence of TE insertions within the *OsPTR* locus, which resulted in a high density of repetitive sequences. The relevant content regarding *OsCACT* has been added to lines 432-436 of the main text.

Supplementary Figure 27. Representative images of the *OsCACT* overexpression lines and their corresponding control of HHZ after 5 days of cold treatment followed by 3 days of recovery at the four-leaf stages. a, Expression levels of *OsCACT* in HHZ and overexpression lines. **b**, Phenotypic comparison of HHZ and *OsCACT* overexpression lines before and after cold treatment. **c**, Box plot showing survival rates of HHZ and *OsCACT* overexpression lines before and after cold treatment.

5. Since the two novel cold-tolerant genes have different allele, it is better using rice NILs material test the cold-tolerant function.

Response: We appreciate your insightful suggestion. We agree that using rice near-isogenic lines (NILs) is an effective approach to investigate the functional differences between alleles of cold-tolerant genes. In this study, we employed a chromosome segment substitution line (CSSL), SN122, which carries a genomic fragment from the *indica* cultivar Nona Bokra (N) in the genetic background of the *japonica* cultivar Koshihikari (K), including the loci of *OsCACT* and *OsPTR*. The genetic background of SN122 is nearly identical to its recurrent parent K, except for the substituted segment, rendering it functionally similar to a NIL. Therefore, our evaluation of cold tolerance in SN122 effectively reflects the allele-specific effects of *OsCACT* and *OsPTR*.

SN122 contains an approximately 0.91 Mb genomic fragment from N located on chromosome 10 (position 22,719,475 to 23,623,527), defined by the markers J02077 and J04003 (**Figure R6a**), within which the N-derived *OsCACT* and *OsPTR* alleles are situated. To confirm the presence of this N-derived genomic region in SN122, we used the marker primers J02071, J02077, J04003, and J03089 (**Table R3**). Genomic DNA (gDNA) was extracted from K, N, and SN122, and PCR products were separated on a 3% agarose gel for 1.5 hours. The results showed that the PCR bands amplified from

SN122 using primers J02077 and J04003 matched those from **N**, while the bands amplified using J02071 and J03089 were identical to those from **K**. These results indicate that SN122 harbors a chromosome segment from J02077 to J04003 derived from **N**, which contains both *OsCACT* and *OsPTR* (**Figure R6b**).

We then evaluated the cold tolerance phenotype of SN122. Compared with its recurrent parent **K**, SN122 showed increased sensitivity to cold stress (**Figure R6c, d**). Consistently, SN122 had significantly higher malondialdehyde (MDA) levels under cold stress, indicating greater membrane damage (**Figure R6e**). These findings suggest that introducing the **N** allele of *OsCACT* and *OsPTR* into a *japonica* background compromises cold tolerance.

Figure R6. Introgression of *indica* alleles of *OsCACT* and *OsPTR* compromises cold tolerance in *japonica* rice. **a**, Genomic fragment containing the *OsCACT* and *OsPTR* allele derived from Nona Bokra (N) on chromosome 10 of SN122. Grey

segments represent the Koshihikari (K) genomic background; black segments denote the substituted N region. The red line and blue line indicate the position of *OsCACT* and *OsPTR*. **b**, Verification of the introduced N fragment in the SN122 genome using specific marker primers. **c**, Representative images of SN122 and K after cold treatment at 4–6 °C for 4 days followed by 3 days of recovery. Scale bar: 5 cm. **d**, Survival rate of SN122 and K after 4 days of cold treatment. Statistical analysis was performed using the Student’s test ($***P < 0.001$). **e**, Malondialdehyde (MDA) content of SN122 and K after 4 days of cold treatment. Statistical analysis was performed using the two-way ANOVA ($***P < 0.001$).

Table R3. Marker primers for amplification of gDNA

Primer ID	Chromosome	Position	Forward primer sequence	Reverse primer sequence
J02071	10	22,438,788	TGCACA ACTACT TGGAACAA	TCTATTCTGAATG ACCAGCC
J02077	10	22,719,475	AGCTTGGTACTG ATTTCGTC	GGCTCAATTTTG AGTGTCT
J04003	10	23,623,527	ATCATGTCAGTA CTACTATG	GTTTGATGTGTA ATTATTGC
J03089	10	23,624,691	TAGGTAGGTTCA CTCTCATC	TTGAAGTAGTGC CTCCATAA

References

- Luo, W. *et al.* COLD6-OSM1 module senses chilling for cold tolerance via 2', 3'-cAMP signaling in rice. *Molecular Cell* **84**, 4224-4238. e4229 (2024). <https://doi.org/10.1016/j.molcel.2024.09.031>
- Klein, S. P. & Anderson, S. N. The evolution and function of transposons in epigenetic regulation in response to the environment. *Current Opinion in Plant Biology* **69**, 102277 (2022). <https://doi.org/10.1016/j.pbi.2022.102277>
- Domínguez, M. *et al.* The impact of transposable elements on tomato diversity. *Nature Communications* **11**, 4058 (2020). <https://doi.org/10.1038/s41467-020-17874-2>
- Wyler, M., Stritt, C., Walser, J.-C., Baroux, C. & Roulin, A. C. Impact of Transposable Elements on Methylation and Gene Expression across Natural Accessions of *Brachypodium distachyon*. *Genome Biology and Evolution* **12**, 1994-2001 (2020). <https://doi.org/10.1093/gbe/evaa180>
- Gill, R. A. *et al.* On the Role of Transposable Elements in the Regulation of Gene Expression and Subgenomic Interactions in Crop Genomes. *Critical Reviews in Plant Sciences* **40**, 157-189 (2021). <https://doi.org/10.1080/07352689.2021.1920731>
- Czajka, K., Mehes-Smith, M. & Nkongolo, K. DNA methylation and histone modifications induced by abiotic stressors in plants. *Genes & Genomics* **44**, 279-297 (2022). <https://doi.org/10.1007/s13258-021-01191-z>
- Ramakrishnan, M. *et al.* Transposable elements in plants: Recent advancements, tools and prospects. *Plant Molecular Biology Reporter* **40**, 628-645 (2022). <https://doi.org/10.1007/s11105-022-01342-w>

- 8 Alonge, M. *et al.* Major Impacts of Widespread Structural Variation on Gene Expression and Crop Improvement in Tomato. *Cell* **182**, 145-161.e123 (2020). <https://doi.org/10.1016/j.cell.2020.05.021>
- 9 Della Coletta, R., Qiu, Y., Ou, S., Hufford, M. B. & Hirsch, C. N. How the pan-genome is changing crop genomics and improvement. *Genome Biology* **22**, 3 (2021). <https://doi.org/10.1186/s13059-020-02224-8>
- 10 Groza, C., Chen, X., Wheeler, T. J., Bourque, G. & Goubert, C. A unified framework to analyze transposable element insertion polymorphisms using graph genomes. *Nature Communications* **15**, 8915 (2024). <https://doi.org/10.1038/s41467-024-53294-2>
- 11 Liu, Z. *et al.* Grapevine pangenome facilitates trait genetics and genomic breeding. *Nature Genetics* **56**, 2804-2814 (2024). <https://doi.org/10.1038/s41588-024-01967-5>
- 12 Mishra, S., Srivastava, A. K., Khan, A. W., Tran, L.-S. P. & Nguyen, H. T. The era of panomics-driven gene discovery in plants. *Trends in Plant Science* **29**, 995-1005 (2024). <https://doi.org/10.1016/j.tplants.2024.03.007>
- 13 Qin, P. *et al.* Pan-genome analysis of 33 genetically diverse rice accessions reveals hidden genomic variations. *Cell* **184**, 3542-3558.e3516 (2021). <https://doi.org/10.1016/j.cell.2021.04.046>
- 14 Shang, L. *et al.* A super pan-genomic landscape of rice. *Cell Research* **32**, 878-896 (2022). <https://doi.org/10.1038/s41422-022-00685-z>
- 15 Kang, Y.-J. *et al.* CPC2: a fast and accurate coding potential calculator based on sequence intrinsic features. *Nucleic Acids Research* **45**, W12-W16 (2017). <https://doi.org/10.1093/nar/gkx428>
- 16 Li, A., Zhang, J. & Zhou, Z. PLEK: a tool for predicting long non-coding RNAs and messenger RNAs based on an improved k-mer scheme. *BMC Bioinformatics* **15**, 311 (2014). <https://doi.org/10.1186/1471-2105-15-311>
- 17 Singh, U., Khemka, N., Rajkumar, M. S., Garg, R. & Jain, M. PLncPRO for prediction of long non-coding RNAs (lncRNAs) in plants and its application for discovery of abiotic stress-responsive lncRNAs in rice and chickpea. *Nucleic Acids Research* **45**, e183-e183 (2017). <https://doi.org/10.1093/nar/gkx866>
- 18 Negri, T. d. C. *et al.* Pattern recognition analysis on long noncoding RNAs: a tool for prediction in plants. *Briefings in Bioinformatics* **20**, 682-689 (2018). <https://doi.org/10.1093/bib/bby034>
- 19 Fueyo, R., Judd, J., Feschotte, C. & Wysocka, J. Roles of transposable elements in the regulation of mammalian transcription. *Nature Reviews Molecular Cell Biology* **23**, 481-497 (2022). <https://doi.org/10.1038/s41580-022-00457-y>
- 20 Schober, P., Boer, C. & Schwarte, L. A. Correlation Coefficients: Appropriate Use and Interpretation. *Anesthesia & Analgesia* **126**, 1763-1768 (2018). <https://doi.org/10.1213/ane.0000000000002864>
- 21 Aoki, K., Ogata, Y. & Shibata, D. Approaches for Extracting Practical Information from Gene Co-expression Networks in Plant Biology. *Plant and Cell Physiology* **48**, 381-390 (2007). <https://doi.org/10.1093/pcp/pcm013>
- 22 Li, J. *et al.* LncTar: a tool for predicting the RNA targets of long noncoding RNAs. *Briefings in Bioinformatics* **16**, 806-812 (2014). <https://doi.org/10.1093/bib/bbu048>
- 23 Leng, Y. *et al.* Genome-wide lncRNAs identification and association analysis for cold-

- responsive genes at the booting stage in rice (*Oryza sativa* L.). *The Plant Genome* **13**, e20020 (2020). <https://doi.org/https://doi.org/10.1002/tpg2.20020>
- 24 Kang, H. M. *et al.* Variance component model to account for sample structure in genome-wide association studies. *Nature Genetics* **42**, 348-354 (2010). <https://doi.org/10.1038/ng.548>
- 25 Wicker, T. *et al.* A unified classification system for eukaryotic transposable elements. *Nature Reviews Genetics* **8**, 973-982 (2007). <https://doi.org/10.1038/nrg2165>
- 26 Robinson, J. T., Thorvaldsdottir, H., Turner, D. & Mesirov, J. P. igv.js: an embeddable JavaScript implementation of the Integrative Genomics Viewer (IGV). *Bioinformatics* **39** (2022). <https://doi.org/10.1093/bioinformatics/btac830>
- 27 Shang, L. *et al.* A complete assembly of the rice Nipponbare reference genome. *Molecular Plant* **16**, 1232-1236 (2023). <https://doi.org/10.1016/j.molp.2023.08.003>
- 28 Song, J.-M. *et al.* Two gap-free reference genomes and a global view of the centromere architecture in rice. *Molecular Plant* **14**, 1757-1767 (2021). <https://doi.org/10.1016/j.molp.2021.06.018>
- 29 Zheng, Z. *et al.* A sequence-aware merger of genomic structural variations at population scale. *Nature Communications* **15**, 960 (2024). <https://doi.org/10.1038/s41467-024-45244-9>
- 30 Sedlazeck, F. J. *et al.* Accurate detection of complex structural variations using single-molecule sequencing. *Nature Methods* **15**, 461-468 (2018). <https://doi.org/10.1038/s41592-018-0001-7>
- 31 Jiang, T. *et al.* Long-read-based human genomic structural variation detection with cuteSV. *Genome Biology* **21**, 189 (2020). <https://doi.org/10.1186/s13059-020-02107-y>
- 32 Heller, D. & Vingron, M. SVIM: structural variant identification using mapped long reads. *Bioinformatics* **35**, 2907-2915 (2019). <https://doi.org/10.1093/bioinformatics/btz041>
- 33 Li, H., Feng, X. & Chu, C. The design and construction of reference pangenome graphs with minigraph. *Genome Biology* **21**, 265 (2020). <https://doi.org/10.1186/s13059-020-02168-z>
- 34 Pritt, J., Chen, N.-C. & Langmead, B. FORGe: prioritizing variants for graph genomes. *Genome Biology* **19**, 220 (2018). <https://doi.org/10.1186/s13059-018-1595-x>
- 35 Sirén, J. *et al.* Personalized pangenome references. *Nature Methods* **21**, 2017-2023 (2024). <https://doi.org/10.1038/s41592-024-02407-2>
- 36 Zhang, F. *et al.* Long-read sequencing of 111 rice genomes reveals significantly larger pangenomes. *Genome Research* **32**, 853-863 (2022). <https://doi.org/10.1101/gr.276015.121>
- 37 Zhang, Y. *et al.* The telomere-to-telomere gap-free genome of four rice parents reveals SV and PAV patterns in hybrid rice breeding. *Plant Biotechnology Journal* **20**, 1642-1644 (2022). <https://doi.org/10.1111/pbi.13880>
- 38 Zhou, Y. *et al.* A platinum standard pan-genome resource that represents the population structure of Asian rice. *Scientific Data* **7**, 113 (2020). <https://doi.org/10.1038/s41597-020-0438-2>
- 39 Li, Z. *et al.* Natural variation of codon repeats in COL11 endows rice with chilling resilience. *Science Advances* **9**, eabq5506 (2023). <https://doi.org/10.1126/sciadv.abq5506>
- 40 He, F., Zhang, F., Sun, W., Ning, Y. & Wang, G.-L. A Versatile Vector Toolkit for Functional Analysis of Rice Genes. *Rice* **11**, 27 (2018). <https://doi.org/10.1186/s12284-018-0220-7>

REVIEWER COMMENTS

Reviewer #1 (Remarks to the Author):

I am satisfied with changes provided in the new version of the manuscript. There is no file in the code submission list. Zenodo page below not found.

Response: Thank you for your positive feedback and for recognizing the improvements in our revised manuscript. We apologize for the oversight regarding the code submission. The correct Zenodo link for accessing the code is: <https://doi.org/10.5281/zenodo.15573226>. We have updated the manuscript and submission system to include this link accordingly.

Reviewer #1 (Remarks on code availability):

Unable to review.

Reviewer #2 (Remarks to the Author):

After revision, the manuscript has been greatly improved. However, some wording is a bit non-standard and it would be good if the authors can have a native English speaker read the manuscript. Please revise and then publish

Response: We sincerely thank you for the positive evaluation and appreciation of the improvements in our revised manuscript.

In response to your suggestion, we have carefully revised the manuscript to improve the clarity and language quality. A native English speaker with expertise in scientific writing has thoroughly reviewed and edited the entire text. In addition, the manuscript was edited for proper English language, grammar, punctuation, spelling, and overall style by one or more of the highly qualified English speaking editors at Springer Nature Author Services. We believe the language is now more polished and meets the standard for publication.

Reviewer #3 (Remarks to the Author):

The authors provided a point-by-point response to my comments in the revised paper. The manuscript identified 30,316 pTEs data and integrated transcriptome and epigenetic modification data; If the data are visualized and easy to be used by breeders, It is valuable for breeding cold-tolerant rice varieties. Though the authors release a web tool (<https://cbi.gxu.edu.cn/RICEPTEDB/>); they just provided the visualized data of MH63RS3 genome, not 131 published high-quality rice genomes, transcriptome and epigenetic data. The authors should integrate the data on the web to visualized it.

Response: We sincerely thank the reviewer for the positive comments and for recognizing the potential value of our work for breeding cold-tolerant rice varieties.

In response to your valuable suggestion, we have now expanded the content available on the web tool (<https://cbi.gxu.edu.cn/RICEPTEDB/>). Specifically, we have added the pTE insertion data for 131 published high-quality rice genomes, integrated the corresponding transcriptome and epigenetic modification datasets, and developed interactive visualization modules that allow users to explore and compare these multi-omics data across different rice accessions.

These updates will facilitate more convenient data access and interpretation for breeders and researchers alike. The database has been fully updated and is now publicly available.

Some comments:

1. The authors claimed that they made the 131 data through the Rice pTE Database; however, I do not find these data in the web.

Response: Thank you for your valuable comment. We apologize for the confusion regarding the availability of the 131-genome data on the Rice pTE Database. We have now resolved this issue and ensured that the pTE insertion data for all 131 rice genomes are available and fully accessible through the database (<https://cbi.gxu.edu.cn/RICEPTEDB/>).

We sincerely thank you for pointing this out and helping us improve the accessibility and utility of our resource.

2. Fig5b, the *OsPTR* expression was downregulated after cold treatment, however, Supplementary Figure 22, *OsPTR* expression was upregulated after 36 and 72 hours of cold treatment. Why?

Response: Thank you for pointing out this discrepancy. We acknowledge that the RNA-seq and qRT-PCR results for *OsPTR* expression under cold treatment were initially inconsistent. Upon careful re-examination, we identified an issue in the original RNA-seq quantification. Specifically, the presence of a TE within the *OsPTR* locus resulted in multi-mapping and ambiguous alignments during the read mapping process, which affected the accuracy of expression estimates.

It is important to note that different RNA-seq analysis tools vary in how they handle multi-mapped and ambiguously aligned reads, and such differences can significantly influence the quantification of gene expression^{1,2}. Properly addressing these reads remains an open challenge in transcriptomic analysis³. A widely accepted approach is to exclude such reads from downstream quantification to minimize bias³.

Accordingly, we reanalyzed the RNA-seq data by applying stringent filtering criteria to remove multi-mapped reads from the BAM files prior to quantification (**Figure R1 and Figure R2**). This corrected analysis produced expression patterns consistent with our qRT-PCR results, confirming the upregulation of *OsPTR* in response to cold treatment (**Figure R3**). We have revised the relevant figures and corresponding descriptions in the manuscript to reflect these updates.

Figure R1. Visualization of RNA-seq read alignments at the *OsPTR* locus before and after read filtering in the *indica* KO.

Figure R2. Visualization of RNA-seq read alignments at the *OsPTR* locus before and after read filtering in the *japonica* TB.

Figure R3. Gene expression levels of *OsPTR*. **a**, Box plot of *OsPTR* expression levels at different cold treatment time points in 10 rice accessions. Statistical analysis was performed using the Wilcoxon test ($*P < 0.05$). **b**, Box plot of *OsCACT* relative expression levels over time under cold stress. Statistical analysis was performed using the Student's t-test, with 0 day as the reference ($***P < 0.001$, ns: $P > 0.05$).

3. Figure R2b This differential methylation correlates with the observed downregulation of *OsCACT* expression. The increase the methylation of the region between the two dashed lines represents the pTE. The TE methylation reduce the *OsCACT*? The author should provide the information of TE sequence of methylation. The TE is still high methylation if it is inserted the other locus?

Response: Thank you for this insightful question. This question is outstanding, as its significance extends beyond our exploration of single-gene expression regulatory mechanisms. It prompts us to identify a novel class of expression silencing sequence tags, which may still be active in the rice populations. The 90-bp pTE located downstream of *OsCACT* exhibits high levels of DNA methylation ($>80\%$, **Figure R4b**), which is significantly correlated with the downregulation of *OsCACT* expression (**Figure R4a**). Therefore, we hypothesize that TE methylation may contribute to the repression of *OsCACT*^{4,5}. The detailed sequence information of this TE is now provided in **Supplementary Table 13**.

Using thresholds of $>80\%$ coverage and $>80\%$ sequence identity, we identified only two additional copies of this 90-bp TE across the genome. Both copies are annotated as consensus TEs and also show high methylation levels ($>60\%$, **Figure R5**). Notably, one of these copies displays a marked difference in methylation levels between the rice varieties 9311 and Nipponbare, suggesting potential epigenetic variation across accessions.

Moreover, we found that the last 69 bp of this 90-bp TE is more widely distributed in the genome, with a total of 24 copies. This 69-bp fragment may represent a conserved core sequence of the TE. To assess whether this sequence has general methylation patterns, we compared its methylation levels to the genome-wide average and to other MITE/*Stowaway* elements (**Figure R6**). We observed that this 69-bp sequence is consistently more methylated than both the genome-wide average and other MITE/*Stowaway* elements, exhibiting moderate methylation levels ($>50\%$).

These findings suggest that this TE and especially its 69-bp core sequence may serve as a potential epigenetic regulatory element. Although further investigation is needed, our study provides an initial indication that TE methylation may have a broader impact on gene expression.

Figure R4. Gene expression and DNA methylation patterns of *OsCACT*. **a**, Box plot of *OsCACT* expression levels at different cold treatment time points in 10 rice accessions. Statistical analysis was performed using the Wilcoxon test ($***P < 0.001$). **b**, DNA methylation levels before and after cold treatment. The region between the two dashed lines represents the pTE.

Figure R5. DNA methylation and ONT read mapping at two additional copies of the 90-bp TE. a, b DNA methylation levels at the TE locus on chromosome 3 (a) and chromosome 4 (b) before and after cold treatment. The region between the two dashed lines represents the TE. ONT read mappings are shown below the methylation tracks.

Figure R6. Comparison of DNA methylation levels across the genome, MITE/*Stowaway*, and the 69-bp core sequence. Genome represented by the average methylation levels across all 12 chromosomes.

Supplementary Table 13. Sequence and annotation information for the pTE downstream of *OsCACT*.

Chromosome	Start	End	Length (bp)	TE type	Sequence
Chr10	24,937,042	24,937,131	90	MITE/ Stowaway	AGTCAGGGGGTGTGGTTAGAACTAGG GACTTATATTTTTGTGAGAGGAACTA AAGTTTAGCCTCACTTTAGTCCCTCC AACCAAACACCAC

4. Figure 5d and Supplementary Figure 27b, in ZH11 background, in normal condition, the *oscact*-ko lines are higher; however, in HHZ background, the *OsCACT* overexpression lines are more strong. *OsCACT* also affect the seedling development?

Response: Thank you for your valuable comment. To determine whether *OsCACT* influences seedling growth and development, we measured key growth parameters, including seedling height and root length, of *oscact*-ko and *OsCACT*-OE lines in the ZH11 background, as well as *OsCACT*-OE lines in the HHZ background, under normal growth conditions.

In the ZH11 background, we assessed the seedling height and root length of *oscact*-ko and *OsCACT*-OE seedlings at the two- and three-leaf stages. The results showed that the growth performance of all transgenic lines was comparable to that of the wild-type ZH11, with no statistically significant differences in either seedling height or root length (**Figure R7**).

Similarly, in the HHZ background, the *OsCACT*-OE lines showed no significant differences in seedling height or root length compared to the wild-type HHZ at the three- and four-leaf stages (**Figure R8**).

These findings indicate that *OsCACT* does not have a detectable effect on early seedling development under normal growth conditions.

Figure R7. Seedling height and root length of *OsCACT* transgenic lines in the ZH11 background. a, b Comparison of seedling height and root length between *OsCACT* transgenic lines and wild-type ZH11 at the two-leaf (**a**) and three-leaf (**b**) stage. Scale bar: 5 cm. All data are presented as mean \pm standard deviation. Statistical analysis was performed using the Student's t-test.

Figure R8. Seedling height and root length of *OsCACT* transgenic lines in the HHZ background. **a, b** Comparison of seedling height and root length between *OsCACT* transgenic lines and wild-type HHZ at the three-leaf (**a**) and four-leaf (**b**) stage. Scale bar: 5 cm. All data are presented as mean \pm standard deviation. Statistical analysis was performed using the Student's t-test.

References

- 1 Sahraeian, S. M. E. *et al.* Gaining comprehensive biological insight into the transcriptome by performing a broad-spectrum RNA-seq analysis. *Nature Communications* **8**, 59 (2017). <https://doi.org/10.1038/s41467-017-00050-4>
- 2 Robert, C. & Watson, M. Errors in RNA-Seq quantification affect genes of relevance to human disease. *Genome Biology* **16**, 177 (2015). <https://doi.org/10.1186/s13059-015-0734-x>
- 3 Stark, R., Grzelak, M. & Hadfield, J. RNA sequencing: the teenage years. *Nature Reviews Genetics* **20**, 631-656 (2019). <https://doi.org/10.1038/s41576-019-0150-2>
- 4 Czajka, K., Mehes-Smith, M. & Nkongolo, K. DNA methylation and histone modifications induced by abiotic stressors in plants. *Genes & Genomics* **44**, 279-297 (2022). <https://doi.org/10.1007/s13258-021-01191-z>
- 5 Ramakrishnan, M. *et al.* Transposable elements in plants: Recent advancements, tools and

prospects. *Plant Molecular Biology Reporter* **40**, 628-645 (2022).
<https://doi.org/10.1007/s11105-022-01342-w>

REVIEWERS' COMMENTS

Reviewer #1 (Remarks to the Author):

No further comments.

Response: We sincerely thank you for your time and the positive assessment of our work.

Reviewer #3 (Remarks to the Author):

The revised manuscript has been greatly improved. Especially adding more data on the web tool (<https://cbi.gxu.edu.cn/RICEPTEDB/>). It is valuable for breeders and researchers. It meets the standard for publication.

Response: We sincerely thank you for your encouraging feedback. We are pleased that the improvements, particularly the enhancement of the online database, were well received. We are also grateful for your insightful suggestions and constructive comments throughout the review process, which have significantly contributed to improving the quality of our manuscript.